# Discovering Symbolic Cognitive Models from Human and Animal Behavior

**Pablo Samuel Castro**[1] **Nenad Tomasev**[1] **Ankit Anand**[1] **Navodita Sharma**[1] **Rishika Mohanta**[2 3] **Aparna Dev**[2]
**Kuba Perlin**[1] **Siddhant Jain**[1] **Kyle Levin**[1] **Noémi Éltető**[1 4] **Will Dabney**[1] **Alexander Novikov**[1]
**Glenn C Turner**[2] **Maria K Eckstein**[1] **Nathaniel D Daw**[1 5] **Kevin J Miller**[* 1 6] **Kimberly L Stachenfeld**[* 1 7]

## Abstract

Symbolic models play a key role in cognitive science, expressing computationally precise hypotheses about how the brain implements a cognitive process. Identifying an appropriate model typically requires a great deal of effort and ingenuity on the part of a human scientist. Here, we adapt FunSearch (Romera-Paredes et al., 2024), a recently developed tool that uses Large Language Models (LLMs) in an evolutionary algorithm, to automatically discover symbolic cognitive models that accurately capture human and animal behavior. We consider datasets from three species performing a classic reward-learning task that has been the focus of substantial modeling effort, and find that the discovered programs outperform state-of-the-art cognitive models for each. The discovered programs can readily be interpreted as hypotheses about human and animal cognition, instantiating interpretable symbolic learning and decision-making algorithms. Broadly, these results demonstrate the viability of using LLM-powered program synthesis to propose novel scientific hypotheses regarding mechanisms of human and animal cognition.

*Equal contribution [1]Google DeepMind [2]Janelia Farm Research Campus, Howard Hughes Medical Institute, Ashburn, VA, USA [3]Laboratory of Neurophysiology and Behavior, The Rockefeller University, New York, NY, USA [4]Max Planck Institute for Biological Cybernetics, Tübingen, Gernamy [5]Princeton Neuroscience Institute, Princeton University, Princeton, NJ, USA [6]Sainsbury Wellcome Centre, University College London, United Kingdom [7]Center for Theoretical Neuroscience, Columbia University, New York, NY, USA. Correspondence to: Kevin Miller <kevinjmiller@deepmind.com>, Kimberly Stachenfeld <stachenfeld@deepmind.com>.

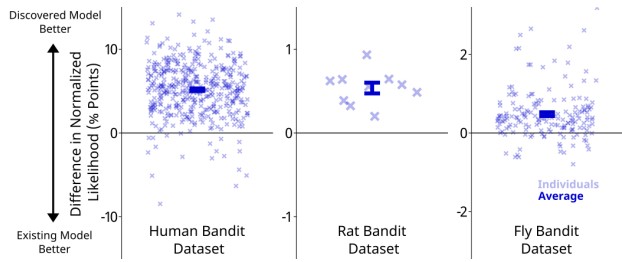

*Figure 1.* **Discovered models outperform human-designed models**. We evaluate the best program discovered by CogFunSearch for each dataset, using average normalized likelihood of the choices made by held-out test subjects, and it to the best existing model from the neuroscience and psychology literature (all $p < 0.002$, signed-rank test) *Left*: Human dataset and model from Eckstein et al. (2024) *Center*: Rat dataset and model from Miller et al. (2021) *Right*: Fruit fly dataset from (Mohanta, 2022; Rajagopalan et al., 2023), model from Ito & Doya (2009).

## 1. Introduction

Symbolic cognitive models are used in neuroscience and psychology to instantiate precise, mechanistic hypotheses about the processes used by the brain to control behavior (Daw et al., 2011; Corrado & Doya, 2007; O'Doherty et al., 2007). These models readily afford interpretation: internal variables like "prediction error" or "forgetting rate" are meaningful in their own right, and suggest possible implementations in the brain. This ease of interpretation has made symbolic models a key tool in understanding how brains produce behavior.

The development of such models has historically been a hypothesis-first process, in which researchers draw on inspiration from the literature and individual human creativity to specify a model, and then refine it to match the idiosyncrasies of behavior. A limitation of this approach is that the space of symbolic models is vast, and it is far from certain that the best possible model for a dataset is one that researchers will have considered (Daw et al., 2011; Wilson & Collins, 2019). Indeed, recent work has used comparisons to flexible recurrent neural networks (RNNs; Dezfouli et al.

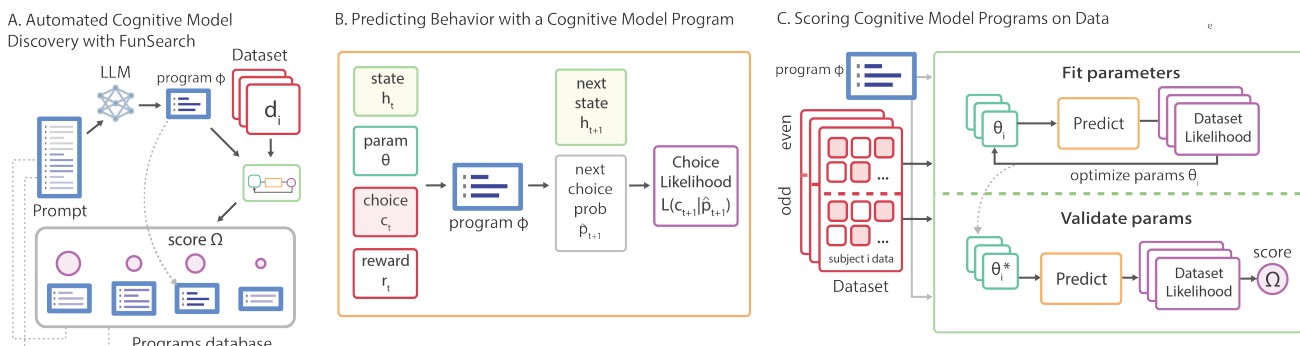

*Figure 2.* **Overview of CogFunSearch**. *A*: CogFunSearch uses LLMs to evolve Python programs that are predictive of a behavioral dataset, and maintains a database of candidate programs which are ranked by their model scores. *B*: Programs $\phi$ receive as input the previous choice $c_t$ and reward $r_t$, an evolving hidden state $h_t$, and a set of trained parameters $\theta$. They output predictive probabilities over the next choice and an updated hidden state. *C*: The model score for programs is computed by fitting parameters on a subset of the dataset and evaluating its normalized likelihood on a separate validation set.

2019b; Song et al. 2021; Ger et al. 2023; Eckstein et al. 2024) to show that commonly-used symbolic models tend to dramatically underfit their datasets.

An appealing alternative is to adopt a data-driven approach, considering a very large space of possible models and allowing quality-of-fit to guide model selection (Brunton & Beyeler, 2019; Miller et al., 2021). As applied to human and animal behavior, this approach has been most broadly successful with model spaces that are differentiable, such as neural networks (Dezfouli et al., 2019a; Miller et al., 2023).

In this work, we demonstrate a method which is able to automatically identify symbolic models that are consistent with a given dataset. We build on advances in program synthesis, a set of methods for automatically identifying computer programs that optimize some objective function. Specifically, we adapt FunSearch (Romera-Paredes et al., 2024), a recent tool that uses Large Language Models (LLMs) to mutate programs within an evolutionary optimization process. Our approach, which we term "CogFunSearch", augments Fun-Search with an additional level of optimization. In the outer optimization loop, FunSearch evolves programs (Fig. 2A), while in the inner loop, model parameters are fit to data (Fig. 2C). We apply CogFunSearch to datasets from three species (humans, rats and fruit flies) performing a classic reward-guided decision-making task which has been the focus of substantial human cognitive modeling effort (Fig. 1; Miller et al. 2021; Eckstein et al. 2024; Mohanta 2022; Rajagopalan et al. 2023). Our discovered programs reliably outperform the best human-discovered cognitive models of which we are aware.

Because CogFunSearch uses LLMs for program generation, it is able to use human-provided information in the prompt and seed program, with more informative prompts resulting

in higher-scoring and more interpretable discovered programs. Discovered programs are often surprisingly readable, for example containing informative variable names and comments. Several unexpected and intriguing motifs are apparent: complex exploration strategies, unconventional value updates, and idiosyncratic patterns of reward-independent choice sequences. Each run of CogFunSearch generates a large number of programs, which show a clear tradeoff between quantitative performance and program complexity. While the highest-scoring programs are more complex than our baseline models, it is possible to identify models for each dataset that are both higher-performing as well as simpler. Broadly, these results validate the use of LLM-based program synthesis for data-driven discovery of cognitive models, and suggest novel hypotheses about reward-guided learning in humans and animals.

## 2. Related Work

**Data-driven Cognitive Modeling** A number of papers have attempted to invert the traditional theory-first approach to cognitive model building and adopt a data-driven approach. Some of these use statistical tools to identify patterns in behavioral data. These include explicitly characterizing patterns in choice and reward sequences using statistical models (Lau & Glimcher, 2005; Sugrue et al., 2004; Ito & Doya, 2009), as well as adding additional degrees of freedom to classic cognitive models (Le et al., 2023; Venditto et al., 2024; Roy et al., 2021). One recent paper has attempted a process of successive model reduction, beginning with flexible statistical models and ending with a model that can be interpreted as a cognitive model (Miller et al., 2021). Related work has applied methods for symbolic regression (Brunton et al., 2016; Landajuela et al., 2022) to discover equations describing behavior. One of these

(Musslick, 2021) considered synthetic data in a variety of decision-making tasks. Another (LaFollette et al., 2023) modeled human behavior in an RL task, but, due to limitations of the technique, did not model choices directly but instead asked subjects to report their internal value estimates and modeled the dynamics of these reports.

**Interpretable Deep Learning** Building on the observation that neural networks often fit data better than symbolic models, several recent papers have introduced methods that attempt to add constraints to deep learning methods to render them more readily interpretable for the purposes of cognitive modeling. One approach is to integrate neural networks into a theoretically-motivated cognitive architecture (Peterson et al., 2021; Eckstein et al., 2024). Another is to constrain information flow within the network, either using explicit information bottlenecks (Miller et al., 2023) or by simply using a very small network (Ji-An et al., 2023). Like this work, our approach optimizes models directly to data, considering a large model space with structural constraints that we hope will afford interpretability. Importantly, unlike network weights, the model space we search (Python programs) is explicitly symbolic, thereby facilitating analysis and interpretation by humans or indeed LLMs.

**Program Synthesis for statistical and agent modeling.** A large body of previous work explores evolutionary methods for discovering computer programs that optimize some objective. A family of work in machine learning uses approaches like these to discover useful algorithms (Real et al., 2020; Co-Reyes et al., 2021; Ellis et al., 2023; Chen et al., 2024). Within cognitive science, programs have previously been proposed as the representational format for learning and storing complex sequences (Planton et al., 2021), geometric shapes (Sablé-Meyer et al., 2022), concepts (Lake et al., 2015), rules (Rule et al., 2024) and behavioral strategies (Correa et al., 2024). This body of work highlights that program synthesis can yield programs that show qualitative commonalities with the average human participants' response patterns. However, to our knowledge, the current work is the first to synthesize programs that maximize the *predictive fit* of individuals' behavior. Perhaps most similar to our methodology, Li et al. (2024) and Shojaee et al. 2025 use LLMs to propose statistical models, then fit the parameters of these models to data.

## 3. Methods

### 3.1. Datasets and Human-Discovered Baseline Models

We consider datasets from humans and other species performing reward-learning tasks in which the subject selects repeatedly between several discrete actions and receives a reward whose magnitude and probability depends only on the chosen action and on a set of dynamic environment parameters that are independent of choice. Such tasks are often called "Dynamic Multiarmed Bandit Tasks", and have been the focus of a great deal of computational modeling effort (Corrado & Doya, 2007; O'Doherty et al., 2007; Daw et al., 2011; Wilson & Collins, 2019). We selected diverse datasets from three different species—humans, rats, and fruit flies—which share two important properties. First, each dataset is unusually large, consisting of many individuals, behavioral sessions, or both. This reduces the risk of overfitting that is inherent in considering a large and flexible space of possible models. Second, each dataset has been the focus of past computational modeling efforts, providing a strong human-discovered baseline against which to compare our discovered models.

**Human Dataset** (Fig. 3A; Eckstein et al. 2024) considers human participants performing a four-alternative task with graded rewards. Participants performed the task online, and indicated their choice on each trial by pressing either 'D', 'F', 'J', or 'K' on their keyboard. Reward was indicated by displaying an integer number of 'points' between 0 and 100, which subjects were asked to maximize. Available rewards followed independent bounded random walks with additional trial-unique noise. Each participant performed up to five back-to-back sessions of up to 150 trials each. The dataset contains choices from 862 participants performing 4,134 total sessions and 617,871 total trials.

Eckstein et al. (2024) performed an extensive comparison of a wide variety of computational cognitive models on this dataset. The model that performed best in this comparison was one we refer to as "Perseverative Forgetting Q-Learning" (PFQ, see Appendix D and Appendix E for details). We adapt PFQ as the human-discovered baseline model for the human bandit dataset.

**Rat Dataset** (Fig. 3B; Miller et al. 2021) considers rats performing a two-armed bandit task with binary rewards. Rats indicated their choice on each trial by entering one of two available nose ports, which were equipped to deliver small liquid rewards. Reward probabilities followed independent bounded random walks. Rats performed daily sessions of approximately one hour. The dataset contains choices from 20 rats performing 1,946 total sessions and 1,087,140 total trials.

Miller et al. (2021) performed an intensive human process of data-driven model discovery on this dataset. This process began with a set of highly flexible models and several iterations of successive model reduction: identifying patterns in the fit parameters of a more-flexible model, and proposing a less-flexible model that embeds those patterns into its structural assumptions. This resulted in a model that we refer to as "Reward-Seeking/Habit/Gambler-Fallacy" (RHG,

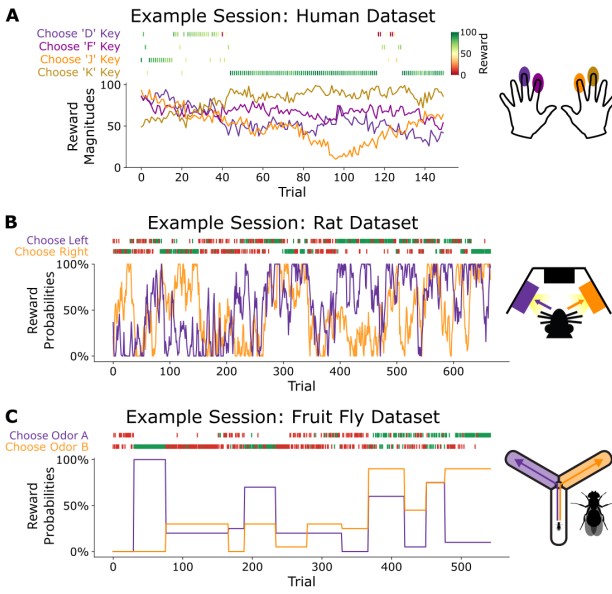

*Figure 3.* **Illustration of datasets** Example behavioral sessions, showing choices and rewards received, as well as reward contingencies, from **A** the Human Dataset, **B** the Rat Dataset, **C** the Fruit Fly Dataset (bottom).

see Appendix D and Appendix E for details). Miller et al. (2021) also performed model comparison between the RHG model and a wide variety of alternative models from the literature, and confirmed that it provides a better fit to data. We adopt RHG as the human-discovered baseline model for the rat dataset.

**Fruit Fly Dataset** (Fig. 3C; Mohanta 2022) considers fruit flies performing a two-armed bandit task with binary rewards. Flies performed the task in a three-armed "Y-maze" setup in which separate odors could be delivered to each of the three arms (Rajagopalan et al., 2023). Flies indicated their choice on each trial by selecting an odor and walking to the end of the associated arm. Reward was delivered via a brief pulse of red light which activated flies' sugar-sensing neurons (Haberkern et al., 2019). Reward probabilities followed a random block structure with randomly sampled reward probabilities and block lengths. Each fruit fly performed one behavioral session. The dataset contains choices from 347 flies performing 68,000 total trials.

Rajagopalan et al. (2023) and Mohanta (2022) have performed extensive model comparison on similar datasets, and identified a popular model known as "Differential Forgetting Q-Learning" (DFQ; Ito & Doya 2009; see Appendix D and Appendix E for details) as performing at least as well as any other. We adopt DFQ as the human-discovered baseline model for the fruit fly dataset.

## 3.2. Problem Formulation

We focus on developing models that predict the choice made by the subject on each trial, using information about the previous choices made and rewards received. We formalize this as follows: On each **trial**, the subject selects between $n$ options and receives a reward. Each subject (indexed $i$) performs one or more sessions ($j$) each consisting of a series of trials ($t$). For each trial $t$ we consider the discrete choice $c \in \{1, ..., n\}$ made by the subject and the reward $r$ received, which may be binary ($r \in \{0, 1\}$, rat and fruit fly datasets) or graded ($r \in [0, 100]$, human dataset). A sequence of $T$ trials constitutes a **session** $s_j := \{(c_1, r_1), \ldots, (c_T, r_T)\}_j$. Conditional on the choice, the delivery of rewards is stochastic according to some experimenter-controlled distribution; in particular, choice $c$'s reward is governed by an independent distribution (Bernoulli or truncated Gaussian, depending on the dataset), with a mean $p_{i,j,t,c}$ that changes over trials according to a dataset-dependent process. For each subject $i$ a set of $N_S$ sessions are collected: $d_i := \{s_{i,1}, \ldots s_{i,N_S}\}_i$, with testing conditions reset between sessions. A **dataset** collects this for $N_D$ subjects: $\mathcal{D} := \{d_1, d_2, \ldots, d_{N_D}\}$. The values of $T$, $N_S$, and $N_D$ will vary based on the dataset in question, as we specify below. See Figure 8 for an illustration of this.

We consider **models** in the form of functions that receive a choice and reward $(c_t, r_t)$ and produce a probability distribution over the next choice $c_{t+1}$, denoted as $\hat{p}_{t+1}$. Models typically also maintain a hidden *state* $h$ that is updated iteratively across a sequence of trials, and are finally parameterized by a vector $\theta$ which is fixed over trials. If we consider the state as an external object $h$ that the model updates, we may formalize a model as $\phi(c_t, r_t, h_t, \theta) \to (\hat{p}_{t+1}, h_{t+1})$.

Following conventions in the literature (which often capture individual differences via a handful of subject-specific parameters, such as learning rate), we assume the model's parameters $\theta$ may vary over subjects, but are fixed over trials and sessions. Thus, for evaluation we optimize the parameters per-subject and score the model by cross-validating across sessions.[1] In particular, for each subject $i$, we split its sessions into even and odd sets $d_i^{\text{even}} := \{s_{i,0}, s_{i,2}, \ldots, s_{i,M-1}\}$ and $d_{i,\text{odd}} := \{s_{i,1}, s_{i,3}, \ldots, s_{i,M}\}$, respectively.

For subject $i$, iterating a model $\phi$ with parameters $\theta_i$ over session $\{(c_{i,0}, r_{i,0}), \ldots, (c_{i,T}, r_{i,T})\}$ will yield the sequence $\{\phi_{\theta_i}(c_{i,0}, r_{i,0}, h_{i,0}) = (\hat{p}_{i,1}, h_{i,1}), \ldots (\hat{p}_{i,T}, h_{i,T})\}$; for each trial $t$ the accuracy of $\phi_{\theta_i}(\cdot)$ is measured via the likelihood of the data under output distribution: $\mathcal{L}(c_{i,t}|\hat{p}_{i,t})$. We use cross-validation to evaluate the overall accuracy of

---

[1]For the fruit fly dataset, since we have only one session per subject, we forego this additional level of variation and treat the dataset as though it were multiple sessions from a single subject with a single $\theta$.

a model as follows. We fit two sets of parameters, $\theta_{i,\text{even}}$ and $\theta_{i,\text{odd}}$, on $d_{i,\text{even}}$ and $d_{i,\text{odd}}$, respectively. All the **even** parameters $\boldsymbol{\theta}_{i,\text{even}}$ are evaluated on the **odd** dataset as:

$$\Omega_i^{\text{even}}(\phi, \boldsymbol{\theta}_i^{\text{even}}) := \exp\left(\frac{2}{MT}\sum_{s\in d_i^{\text{odd}}}\sum_{t=0}^{T}\log\mathcal{L}(c_{i,t}|\hat{p}_{i,t})\right)$$

The score $\Omega_i^{\text{odd}}(\phi, \boldsymbol{\theta}_i^{\text{odd}})$ is computed analogously, and we define the overall subject score $\Omega_i(\phi)$ as the average of the two. The dataset score over the entire dataset $\mathcal{D}$ is computed by averaging over all subject scores in the dataset. See Sec. A and B for further specific details of scoring and how the datasets were divided for cross-validation.

### 3.3. Evolving Cognitive Models With FunSearch

We use the above score function to evolve programs using FunSearch (Romera-Paredes et al., 2024). Briefly, FunSearch uses large-language models (LLMs)[2] to create "mutations" of existing programs. To generate each new program $\phi'$, the LLM is prompted with a pair of existing programs $\phi_0$ and $\phi_1$, along with information about how the programs will be evaluated and a directive to generate an improved program $\phi'$. The new program is then scored using the process described above (Section 3.2) and is added, along with its score, $\Omega(\phi')$, to a program database. To select "parent" programs for each LLM sample, FunSearch stochastically samples programs from this database based on their scores. We refer to the combination of FunSearch with our scoring function and framework for evolving cognitive models as "CogFunSearch" (Figure 2).

Each CogFunSearch experiment is run with three separate seeds, each running for seven days, generating 100,000s to 1,000,000s of programs depending on the dataset. The produced models have up to ten unique parameters, and no bound on the hidden state size.

To validate the discovered models, we hold out an additional test set containing subjects not seen by the FunSearch process. The programs that emerge from CogFunSearch are fit to data from these subjects, and again evaluated using an even-odd split. Unless otherwise indicated, all reported scores are performed on these held out test subjects. More details on how data were structured can be found in Appendix B.

Our optimization thus effectively nests two levels: the outer-level CogFunSearch optimizes over Python programs (Figure 2A), whereas the inner level fits program parameters over a dataset to evaluate performance (Figure 2C). The result is that CogFunSearch produces model "templates" that can be fit on a dataset, capturing across-subject and

within-subject patterns of behavior in an implicitly hierarchical way. To the best of our knowledge, this is a novel application of FunSearch, which has thus far been used as a single-level evolutionary optimization process.

### 3.4. Configuring FunSearch

FunSearch is initialized with a prompt that contains an initial seed program $\phi_0$ as well as documentation about the problem and scoring process. We explored four different initial prompts for CogFunSearch that varied the informativeness of the seed program from mere "hints" as to what the programs should do–in code and in natural language documentation–to more substantive starter code (see Appendix F).

*LowInfo* contains function and variable names (`model`, `x`, `y`) and documentation that describes fitting a generic statistical model. *Structured1* includes informative function and variable names (`agent`, `choice`, `reward`) and documentation describing the data and modeling framework, but no explicit suggestions as to the model's structure. *Structured2* modifies *Structured1* to include hints on how to structure and style the program. Finally, *FullModel* initializes seed program $\phi_0$ to the baseline program itself, permitting a "warm start." These represent different levels of information with which a researcher might approach a modeling problem in different settings.

## 4. Results

We performed three independent runs of CogFunSearch on each of our three datasets (Human, Rat, Fruit Fly) using each of the four seed programs described above (LowInfo, Structured1, Structured2, FullModel; Sec. 3.4). We ran each of these until it had sampled at least 150,000 new programs. This left us with a very large library of evolved programs for each dataset, each of which can in principle be considered a hypothesis about the cognitive mechanisms at play in that dataset. Whether a program makes a good cognitive model is typically evaluated according to two criteria: how well they match data and how complex they are. Below, we consider each of these questions in turn.

### 4.1. Best Discovered Programs Fit Data Well

We first considered the best programs discovered for each dataset and asked how well they perform at fitting data. To identify a single best program for each dataset, we selected the program, from among all CogFunSearch runs across all seed programs, which earned the highest score on the training dataset. For the rat and fruit fly datasets the best performing programs came from runs using the FullModel seed program, and for the human dataset the best performing program came from a run using the Structured2 seed

---

[2]Specifically, we use Gemini 1.5 Flash (Gemini Team, 2024).

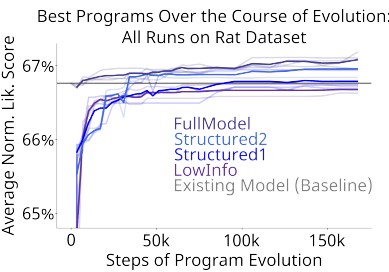

*Figure 4.* **Quality-of-fit over the course of program evolution.** Results averaged over held-out test subjects, computed at each timepoint in evolution for the then-best program of each CogFunSearch run. Individual runs are plotted as light lines, while dark lines represent the average across all runs.

program. The complete code for these "best programs" can be found in Appendix G.

We find that the best discovered programs (maximizing over runs and seed programs) substantially outperform the human-discovered models for each dataset (Wilcoxon signed rank tests on score improvement by subject: $W = 1772$ (Human), $0$ (Rat), $1479$ (Fruit Fly); all with $p < 0.002$). Figure 1 shows relative improved likelihood for each held-out test subject as percentage points between the top-scoring CogFunSearch program (FS) and the species-specific baseline model (BL) ($\Omega_{\text{FS}}^i - \Omega_{\text{BL}}^i$). Across datasets, the majority of test subjects are better fit with the best CogFunSearch program than by the baseline (95.6% for Human, 100% for Rat, and 83.8% for Fruit Fly), with average improvements in normalized likelihood score of $5.17 \pm 0.15$ (Human), $0.54 \pm 0.06$ (Rat), $0.48 \pm 0.05$ (Fruit Fly).

Decomposing these results by seed program type and multiple runs (and aggregating over subjects) to investigate consistency, we found that CogFunSearch yielded well-fit models reliably: in 35 of our 36 runs the highest-scoring

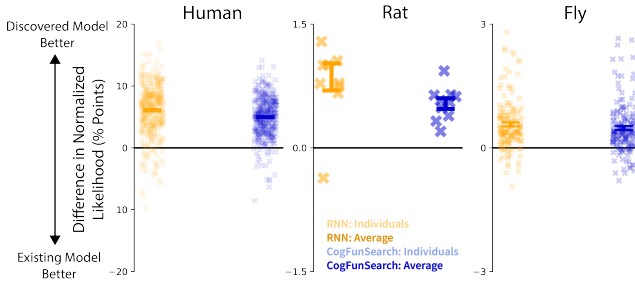

*Figure 5.* **CogFunSearch closes the gap with neural networks.** We compare the difference in normalized likelihood between the RNN and best discovered CogFunSearch programs. Across all species, the RNN narrowly outperforms best CogFunSearch program, although CogFunSearch has bridged the majority of the gap. Scores can be found in Table 1.

program outperformed its baseline model when evaluated on held-out data (Figure 4). The performance of the final programs was broadly similar between runs using the three informative seed programs (Structured1, Structured2, and FullModel), but yielded lower-performing programs with LowInfo (p=0.05, 0.08, 0.05, 0.27, 0.05, 0.05, 0.65, 0.05, and 0.13, for the nine comparisons between LowInfo and the other seed programs for the three species; Wilcoxon rank-sum tests across runs). Runs with different seed programs also differed in how many cycles of evolution were required to identify well-fit models (Fig. 4, rat dataset; Fig. 10, human and fly datasets). These results demonstrate the ability of CogFunSearch to improve upon a human-specified model (FullModel), to synthesize a model that fits well given no information about the specific problem being solved (Low-Info), as well as to make use of intermediate amounts of information (Structured1 and Structured2).

To further test whether the best discovered programs provided a quantitatively good match to data, we compared their fit to held-out data with that of recurrent neural networks. Specifically, we train a GRU model (Cho et al., 2014) over $d_{i,\text{even}}$, run a sweep on the number of hidden units (over $\{1, 2, 4, 8, 16, 32, 64, 128\}$), and use early-stopping to select the best parameters. All the variants were trained with the Adam optimizer (Kingma & Ba, 2015) with a learning rate of $1e - 4$. We found that the performance of the best discovered programs is broadly similar to that of the best RNNs, suggesting that they are capturing nearly all of the structure that exists in our datasets (Figure 5).

A second measure of the quality of a model is its ability to generate synthetic datasets which reproduce the scientifically important features of the real datasets (Wilson & Collins, 2019; Palminteri et al., 2017). To test our best programs, we used them to generate synthetic datasets, matching the real datasets for number of trials per session, number of sessions, and reward contingencies. To characterize patterns present in each dataset, we applied a trial-history logistic regression analysis common in behavioral neuroscience (Lee et al., 2005; Lau & Glimcher, 2005), which quantifies the extent to which rewards are followed (at various trial lags) by repeated choices ("reward-seeking") and the extent to which choices tend to be repeated ("choice perseveration"). The Rat and Fruit Fly dataset exhibit different patterns of reward-seeeking and choice perseveration, and we find that the patterns revealed by this analysis match closely between the synthetic and the real datasets (Fig. 6). This indicates that our discovered programs are able to match in detail the relationship between past choices and rewards, and future choice in our datasets.

A final, very practical feature is that a model should be easy to work with in the sense of having a likelihood landscape that facilitates parameter optimization via gradient descent.

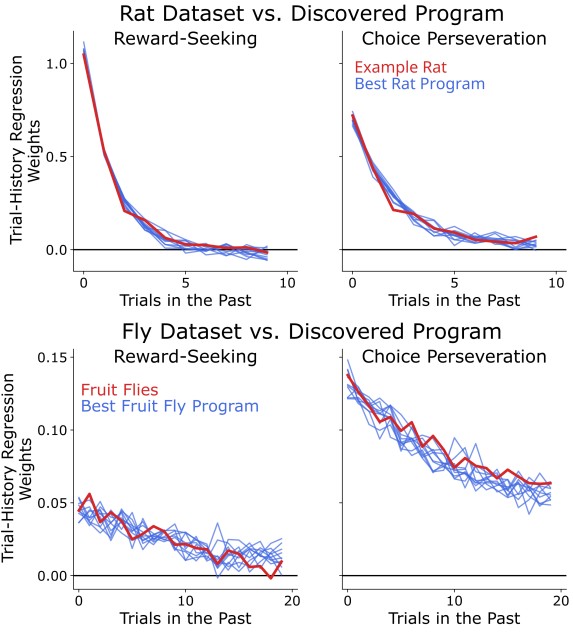

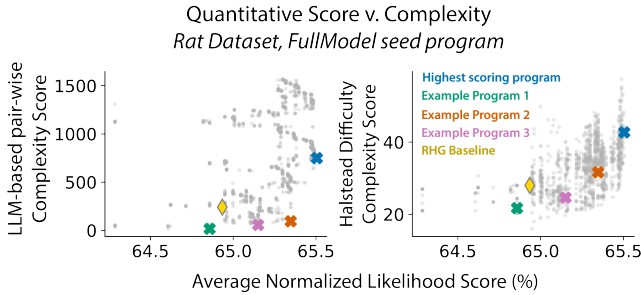

*Figure 7.* **Quality-of-Fit trades off with Complexity.** Two complexity measures (LLM-based, Halstead Difficulty) both show a tradeoff between program complexity and normalized likelihood on training data. See Appendix I for example programs from along the frontier.

*Figure 6.* **Synthetic datasets generated by discovered programs match patterns in real data.** Red lines show lagged regression model fit to behavioral data. Blue lines ("Best Program") show fits to ten synthetic datasets each generated using the best program, model parameters fit to the data, and task conditions matched to the real dataset. Multiple synthetic datasets are shown to illustrate the variability that results from stochasticity in the model.

Human-designed models often lack this feature, and improving it is the target of effortful reparameterization (Wilson & Collins, 2019). We find that our CogFunSearch programs return stable parameter estimates when fit multiple times, indicating that they have favorable likelihood landscapes (Fig. 17). They also find similar parameters across different splits of the data, suggesting stable solutions given the variability in the data (Fig. 18).

### 4.2. Quality-of-Fit Trades Off With Complexity

To evaluate the complexity of our discovered programs, we computed two distinct measures for each. The first is one of the Halstead (1977) measures, used in software engineering to quantify code complexity. We focus on the "difficulty" measure, which relates to how difficult the code is for a human to read and understand, for example during code review. The second is a novel measure based on prompting an LLM to rate the relative complexity of our programs (details in C.3). We have found this measure to align well with our subjective sense of how easy it is to understand these programs. We compute these measures both for the programs returned by CogFunSearch as well as our human-discovered baseline models.

Examining these complexity measures (Figure 7 for Rat Dataset, Structured2 configuration; Figure 12,13 for all other datasets and configurations), we see a clear tradeoff between likelihood score and complexity, with higher scoring programs tending to have higher complexity scores. These plots also allow us to identify example programs on the "efficient frontier", earning the lowest complexity for their likelihood score. Researchers interested in identifying models with different tradeoffs between fit and simplicity might choose to examine programs at different positions along this frontier. Comparing the frontier to the baseline program for each dataset, we see that the CogFunSearch programs include some that are improvements both in quality-of-fit and in complexity (e.g. "example program 3" in Figure 7), and that this is true across most seed programs and datasets (Figures 12 and 13).

## 5. Discovered programs exhibit novel strategies

Despite not having been explicitly optimized for interpretability, we find that the best discovered CogFunSearch programs exhibit intriguing and often interpretable strategies. Moreover, many of the programs involve parameters, learning updates, and decision-making rules that were not present in our baselines. Here we aim to understand what computational motifs characterize our highest-scoring programs (see Appendix G for full programs).

**Top Scoring Human Program** A salient feature of this program was that the bulk of the code—and of the agent's internal state—was devoted to choice history rather than reward tracking. In addition to variables tracking the expected values of the four actions, the best program introduced a number of novel variables that each track different reward-independent statistics of previous choices:

```
q_values = agent_state[:4]
old_choice = agent_state[4]
trial_since_last_switch = agent_state[5]
exploration_rate = agent_state[6]
cumchoice = agent_state[7:11]
```

These variable names are largely indicative of their actual function. ("q values" is standard terminology in reinforcement learning models for tasks of this kind to represent the reward an agent expects to receive following an action e.g. Ito & Doya 2009). Together, these variables can capture behavior motifs on different timescales, such as long runs of the same choice, and other more complicated choice patterns such as cycles or switching within vs. between hands. Such patterns (several of which were also documented by Eckstein et al. 2024) are evidently prominent in four-alternative bandit datasets, and not wholly captured by the simple stay-switch perseverative mechanisms included in the baseline model, which were inherited from models of two-alternative bandit tasks (Lau & Glimcher, 2005). This previously unappreciated aspect of the task may account for the relatively weak performance of baseline models in the human dataset relative to the other species.

Three independent runs with the Structured2 seed program, (highest scoring program for Human), separately discovered a common (but, to our knowledge, novel) motif whereby the learned values were decayed, at each step, toward their average:

```
# Best Human Bandit Program
q_values = (1 - exploration_rate) * q_values + (
    exploration_rate * jnp.mean(q_values))

# Program 2
q_values = (1-bias) * q_values + bias * jnp.mean(q_values)

# Program 3
updated_q_values = (1 - alpha_choice) * updated_q_values + (
    alpha_choice * jnp.mean(q_values))
```

All three of these top programs also introduced a ceiling on the logit-derived choice probabilities, which was not included in the baseline model but has been used in earlier published models of similar tasks (Shteingart et al., 2013). Suggestively, variable names from all three programs referred to this parameter as a "lapse" rate, presumably reflecting the prompted LLM's contextual sensitivity and pre-training on the psychology literature, where that term is commonly used (Green et al., 1966). In the reinforcement learning literature, in contrast, that rule is usually known as "$\epsilon$-greedy".

**Top Scoring Rat Program**     The best rat program was derived from the FullModel seed program. This was the only dataset for which we found the state variables from the baseline model still present in the final program, and supports the notion that RHG (which was derived by a manual data-driven approach, Miller et al. 2021) is a strong

baseline. The additional state variables implement a learning rule similar to those found in "Q-learning" models (e.g. Ito & Doya 2009), updating the value of the chosen action towards the reward, but which also updates it towards the value of the unchosen arm:

```
state_w[choice] = (
    alpha_q * state_w[choice] +
    (1 - alpha_q) * reward_for_update +
    alpha_bias * gamma_w * state_w[1-choice]))
```

(**gamma_w** and **alpha_bias** are constrained to be positive). This mechanism somewhat echoes the decay-toward-average motif discovered above. Other top programs implemented a term tracking recent average reward, which modulated the exploration rate, reminiscent of several existing cognitive models (Aston-Jones & Cohen, 2005; Eldar et al., 2016; Palminteri et al., 2015).

**Top Scoring Fruit Fly Program**     The best Fruit Fly programs were also derived from FullModel. Similar computational motifs from the original baseline were present, but often in altered form. "Forgetting", which decays the value of the unchosen action, was modified to apply only when a positive reward prediction error was present. Differential learning rates for rewards versus omissions were replaced by the differential forgetting of values. This program again exhibited a mixture of exploration strategies that combined biased softmax and $\epsilon$-greedy.

**Subjective Code Quality**     Broadly, we noted that parameters were often assigned names that were meaningful in the context of RL (e.g. **lapse_rate**, **initial_values**). These names were often indicative of their role in the program, and a sense of the program's function could be gleaned by browsing assigned parameters. Moreover, as indicated above, many lines of code were sufficiently comprehensible that they could, with moderate effort, be meaningfully summarized.

However, we note clear room for improvement in terms of readability and complexity. While variable names tended to be informative, for all seed programs besides LowInfo, a number were vague (**bias2**), unnamed (**params[0]**), or misleadingly named (e.g. **exploration_rate** in the best Human program snippet above actually modulated value forgetting). Programs frequently attempted to index more parameters than were available (which does not throw an error in compiled JAX code, but instead defaults to indexing the final array element), and occasionally defined state variables they never used (e.g. the best Rat program defines an 18 dimensional hidden state, and proceeds to update 7 of them). Programs had occasional tautological lines (**agent_state = agent_state**), or terms that seemed meaningful but were in fact multiplied by a parameter that always fit to a value of zero (**gamma_q** in best Rat program). For examples

of programs that strike a different balance of interpretability and performance (as in Figure 7, see Appendix I).

## 6. Conclusion

We introduce *CogFunSearch*, an extension of FunSearch, which enables data-driven discovery of cognitive models of human and animal behavior. We find that CogFunSearch can discover programs that outperform state-of-the-art baselines for predicting animal behavior, while remaining largely interpretable. By sampling different programs from within the database generated by CogFunSearch, we can identify programs with different interpretability / performance trade-offs.

Future work will aim to close the remaining quantitative gap between the discovered programs and RNN, and to improve model interpretability. Hybrid neurosymbolic architectures with evolutionary architecture search and model distillation comprise two promising approaches. While we focus on balancing quality-of-fit with interpretability, future work may also look at trading off quality-of-fit with other desirable model qualities, like runtime, optimizability, data efficiency, or generizability.

An important area for future work is to improve the efficiency of (Cog)FunSearch, which required hundreds of thousands of LLM calls to identify the best programs that we reported here. A simple idea we explored in this direction is to do a form of rejection sampling during program generation, which focused the optimization process on promising programs. Although this approach risks converging to local optima (due to a lack of exploration), our results indicate that we were still able to obtain interpretable programs that outperformed the baseline with only tens of thousand samples. More sophisticated approaches can potentially improve further on this efficiency-accuracy tradeoff. We provide more details in Appendix C.6. It would also be worthwhile to explore the efficacy of other LLM models, other AI discovery methods in the literature, as well as non-differentiable approaches (Acerbi & Ma, 2017; Ma et al., 2024; Zhou et al., 2024; Misra & Kim, 2024; Ye et al., 2023; Lu et al., 2024; Rmus et al., 2025).

Relatedly, the sheer density of programs output by CogFunSearch also highlights that choice predictiveness alone is likely not sufficient to identify a single "best" model. This contrasts against the implicit assumption of much previous work in this area, which has often focused on ranking smaller, curated sets of hand-designed models. An important question for future work will be systematically exploring similarities and differences between groups of models with similar predictive power: to what extent are they different expressions of the same input-output relationship, or notational variants that reparamaterize equivalent param-

eters or hidden states? Further LLM-based analysis (like the first steps we take with interpretability ranking) may help to address these questions. Also, to the extent different expressions of a model are behaviorally equivalent in some regime, they might suggest additional experimental tests or measurably different neural implementations.

This work demonstrates the potential of LLM-guided program synthesis for discovering novel models of human and animal behavior. There are exciting opportunities to apply this to other, potentially more complex behaviors and cognitive processes. More broadly, these results provide a promising indication that LLM-guided discovery can be a generally useful scientific tool in finding predictive models that are interpretable as well as predictive.

## Impact Statement

This work introduces an AI tool for discovering interpretable predictive models of human and animal cognition. Models that explain behavior and can offer hypotheses about underlying mechanisms are useful for psychologists and neuroscientists, and tools for improving this practice stand to advance the field considerably. Long-term, tools for building better predictive models of behavior and cognition also stand to have a positive impact in the clinical psychology setting, as they might support diagnosis or intervention.

Modeling human behavior is also associated with a degree of risk, as models that effectively describe and predict behavior could also be deployed to disrupt human behavior in targeted, adversarial ways. Dezfouli et al. (2020) illustrate how predictive models of learning and decision-making can be used to bias decision-making, and a similar risk applies to our setting as well.

An additional consideration is the compute cost associated with using LLMs. In discovering these programs, CogFunSearch placed hundreds of thousands of LLM calls per run. We explore a potentially more efficient approach in Appendix C.6; however, this remains a limitation of LLM-based methods.

**Acknowledgements** We would like to thank Doina Precup, Joelle Barral, Zoe Ashwood, the FunSearch community at Google DeepMind (Romera-Paredes et al., 2024; Novikov et al., 2025), and the rest of Google DeepMind for providing support and feedback during the preparation of this work.

We would also like to thank the anonymous reviewers for their useful suggestions for improvement. Finally, the authors would also like to thank the Python community (Van Rossum & Drake Jr, 1995; Oliphant, 2007) for developing tools that enabled this work, including NumPy (Harris et al., 2020), Matplotlib (Hunter, 2007), Jupyter (Kluyver

et al., 2016), Pandas (McKinney, 2013) and JAX (Bradbury et al., 2018b).

The work at Janelia Reseach Campus was supported by funding from the Howard Hughes Medical Institute. We thank Kaitlyn N. Boone (Janelia Project Technical Resources) for assistance with fly behavior assays.

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

# A. Extended Methods

## A.1. Evaluation

Our models are discovered using a bilevel optimization procedure in which programs are optimized *across-subject* using an LLM-powered evolutionary algorithm in the outer loop, and parameters of the model are fit *per-subject* using gradient descent in the inner loop (Fig. 8). This necessitates two levels of training and validation test sets: one (for the outer loop) that splits the data into train and test subjects $\mathcal{D}_{\text{train}}$ and $\mathcal{D}_{\text{test}}$ and one that splits the data of subject $i$ into sessions $d_{i,\text{odd}}$ and $d_{i,\text{even}}$ used for crossvalidation. For clarity, we refer to validation at the level of the outer loop as "evaluation" and validation at the level of the inner loop as "scoring."

In Sec. 3.2, we describe the procedure by which each program is scored on data from each subject using two-fold cross validation across sessions. For each subject $i$, we divide sessions into equal splits (even and odd sessions), fit the parameters of the model $\phi$ separately to each split, and evaluate the fit parameters $\boldsymbol{\theta}_{i,\text{even}}$ and $\boldsymbol{\theta}_{i,\text{odd}}$ on data from the opposite split. This gives us two scores per subject: $\Omega_{i,\text{even}}(\phi, \boldsymbol{\theta}_{i,\text{odd}})$ and $\Omega_{i,\text{odd}}(\phi, \boldsymbol{\theta}_{i,\text{even}})$ – the likelihood of data from even sessions given parameters fit to odd data, and vice versa. We average these scores together to produce a single score $\Omega_i(\phi)$ per subject. These per-subject scores are then aggregated across the data in the training by computing the average across all subjects in the dataset $\Omega(\phi)$.

During scoring, parameters for the program are fit to the training data with gradient descent. CogFunSearch's programs must be implemented in Jax (Bradbury et al., 2018a) so that they are differentiable. This is communicated to the LLM by including type hints and Jax imports in CogFunSearch's specification. Non-jittable functions will usually throw an error and not get added to the program database.

We use the AdaBelief optimizer with learning rate $5 \times 10^{-2}$ which is run until convergence or until 10,000 steps of gradient descent are reached. In order to test convergence, we compare the current score at iteration $k$, $\Omega_k$ to the previously recorded score $\Omega_{k-100}$ every 100 steps. If the relative change in score $|(\Omega_k - \Omega_{k-100})/\Omega_{k-100}|$ is less than a convergence threshold $10^{-2}$, we conclude that parameter fitting has converged. In order to protect against local minima, this parameter fitting process is repeated from different initial parameters up to 10 times, or until 3 programs have converged to the current best.

We maintain a group of held-out subjects, $D_{\text{test}}$, in order to validate our discovered programs. Data from these subjects was not seen at all during the evolution or scoring of FunSearch programs. In order to score programs on this held out data, parameters are fit to data from each subject using the cross validation procedure described above.

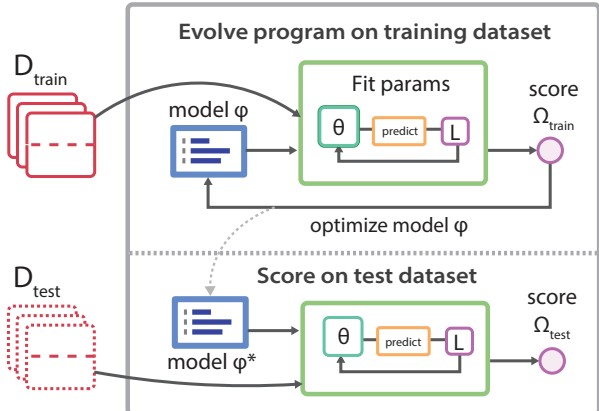

**Bilevel optimization**
Finding programs across-subject and fitting programs within-subject

*Figure 8.* **Organizing data for train and test.** The full dataset $D$ is split into $D_{\text{train}}$ and $D_{\text{test}}$; each of these is further split into fit and eval subsets (i.e. $D_{\text{train}}^{\text{fit}}$ and $D_{\text{train}}^{\text{eval}}$). fit and eval will correspond to even and odd sessions, respectively, and then parameter fitting will be repeated and the scores averaged. In the top panel, generated programs are evaluated by first fitting params on $D_{\text{train}}^{\text{fit}}$ and then evaluating them on $D_{\text{train}}^{\text{eval}}$. After the evolutionary process has completed, the resulting program is tested on $D_{\text{test}}$ in an analogous manner. This latter score is what is used for our comparisons throughout the paper.

# B. Extended Dataset descriptions

Generally, we divided all of our datasets into training and test sets to validate programs, which were then further divided into even and odd datasets to validate parameters. However, the Human, Rat, and Fruit Fly datasets differed in key structural ways which affected the data partitioning.

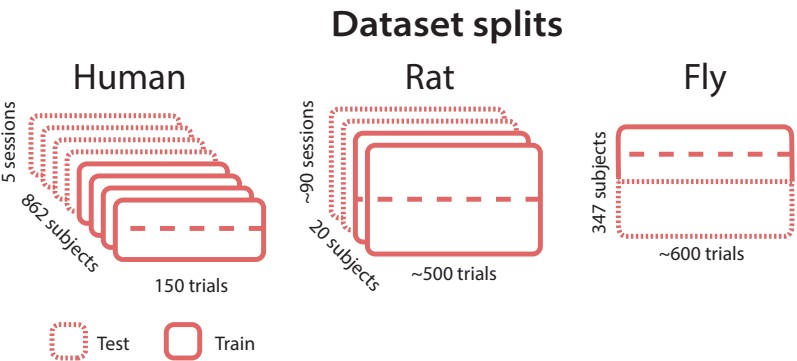

*Figure 9.* **Organizing data for train and test.** In Human and Rat datasets we use half of the subjects for training, and half for testing; for each train subject, we use half of its sessions for parameter fitting and half for evaluation. For the Fly dataset we use half the subjects for training and half for testing, and proceed similarly as for the other datasets.

As detailed in Fig. 9, the Human dataset contained a large number of subjects and trials, but only five sessions per subject. Despite the small number of sessions, we found we could fit the ten parameters of CogFunSearch programs (using cross-validation given the data splits denoted by the horizontal dashed line in Fig. 9), and programs were optimized using scores aggregated across subject (validating on the test subjects denoted with the dotted lines). The four parameters of the PFQ baseline were fit similarly.

However, the small number of sessions meant that attempting to fit the parameters of an RNN to each subject resulted in extremely low performance. Thus, as described in the main text, we followed the approach of Eckstein et al. (2024) and fit an across-subject RNN model, joining sessions from each subject to form one long concatenated session. While parameters are therefore not adapted per-subject for this model, the RNN can adapt in its latent space to reflect subject-specific activity. The data organization that we used for fitting RNNs ignored the per-subject division into even/odd splits (denoted by a dashed line in Fig. 9), and only required the train/test split indicated at the across-subject level (like the division illustrated for Fly). While is not an apples-to-apples comparison, it does compare each model in the regime where they are best set up to succeed.

The Rat dataset contained an intermediate number of subjects, but with far more sessions and trials per session, permitting per-subject parameter fitting for all models. An important property of the experiment design is that 20% of trials were randomly selected to be "instructed", meaning that the animal was instructed to make a particular choice rather than allowed to choose freely. Failure to make the instructed choice resulted in adverse outcomes (loud noise and a timeout). This was included to enforce exploration, and ensure that the animal is receiving informative experience to drive the learning process that is the targetted object of study. This is similar to "teacher forcing", which is commonly used for training sequential models.

The fly dataset contained a large number of subjects, but contained only one session per subject. This precluded any possibility of fitting parameters per-subject; rather, we followed prior work (Mohanta, 2022; Rajagopalan et al., 2023) and fit parameters of a metasubject model for CogFunSearch, DFQ baseline, and the GRU. The way we split the data is different for this data also differed (Fig. 9): rather than having train and test subjects that were divided into sessions, we had train and test subject populations and only the training subjects were further subdivided into even / odd splits. Model evaluation is then performed by fitting the parameters of the model to all subjects in the training set, and then reporting performance on the held-out test subjects.

A nuance about the Human dataset is that what we refer to as "sessions", for notational consistency , would more accurately

be referred to as blocks, because they are experienced by the subjects consecutively. Between blocks, subjects received summary feedback about the previous session and took a short break. The appearance of arms changed after each block and the drifts reset. Because of the temporal proximity of sessions in this dataset, there are likely dependencies between these sessions that we are not explicitly capturing.

## C. Extended Results

### C.1. Normalized likelihoods

Table 1 presents the average absolute likelihoods for the Baseline, RNN, and CogFunSearch methods, with standard errors.

|  | **Baseline** | **RNN** | **CogFunSearch** |
|---|---|---|---|
| **Human** | $55.77 \pm 0.64$ | $62.04 \pm 0.62$ | $60.93 \pm 0.63$ |
| **Rat** | $66.63 \pm 0.53$ | $67.41 \pm 1.21$ | $67.22 \pm 1.25$ |
| **Fruit Fly** | $53.99 \pm 0.24$ | $54.71 \pm 0.57$ | $54.62 \pm 0.57$ |

*Table 1.* Normalized likelihood scores (as percentages) and standard error, as computed over held-out test subjects, for the evaluated methods on each of the datasets.

The number of samples from the LLM required by CogFunSearch to produce these models was 1.2M (Human), 300K (Rat), and 200K (Fly).

### C.2. Differences across Seed Program

Here, we present extended results on the performance differences induced by the different seed programs considered throughout training. Figure 10 shows how performance of the best discovered program changes over the first 150,000 LLM samples for each CogFunSearch (fruit fly and human datasets, corresponding plot for the rat dataset can be found in the main text as Figure 4). Figure 11 shows the performance of the best program discovered at the end of evolution for each CogFunSearch run. Performance is often better than in Figures 11 and 10 as many CogFunSearch runs were allowed to proceed for more steps of evolution.

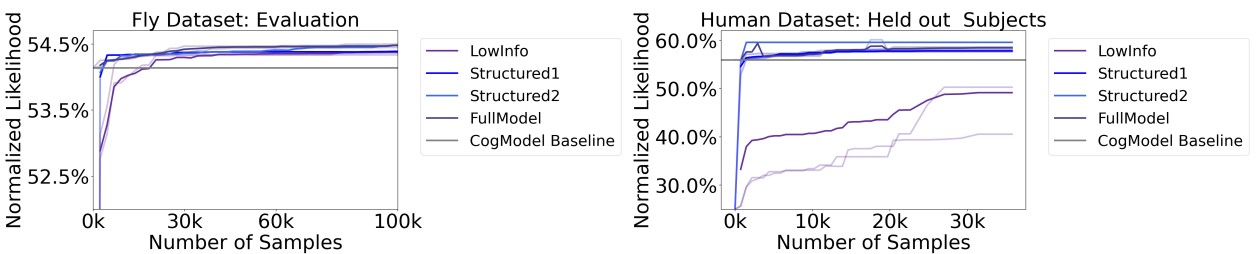

*Figure 10.* **Impact of seed program on quality and discoverability of models.** Evaluation results of (left) fly dataset and on (right) human dataset, computed at each timepoint in evolution for the best program so far discovered for each CogFunSearch run. Individual runs are plotted as light lines, while dark lines represent the average across all runs.

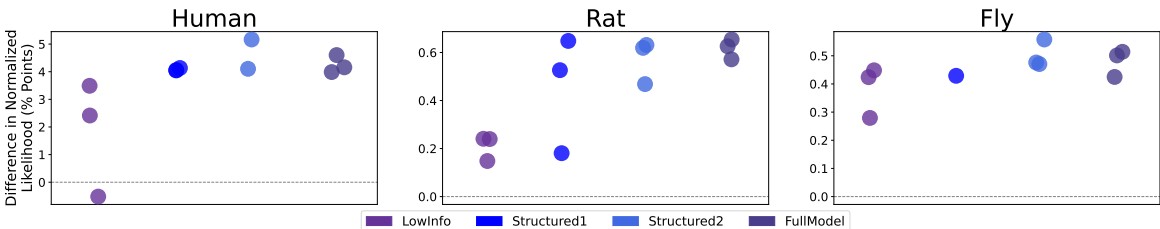

*Figure 11.* **Impact of seed program on final model performance,** from different seeds and independent runs, for each of the datasets. Each dot represents the average normalized likelihood improvement (over baseline) averaged over the held-out test subjects.Each dot represents an individual run, illustrating the variance incurred from independent runs.

## C.3. Complexity Statistics

We evaluate the interpretability of the generated programs as well as the baselines, relative to their scores. To do so, we use one of the Halstead measures (Halstead, 1977), which is a standard metric for software complexity. Figure 12 plots, for each dataset, the generated programs from different program seeds, as well as the respective baseline.

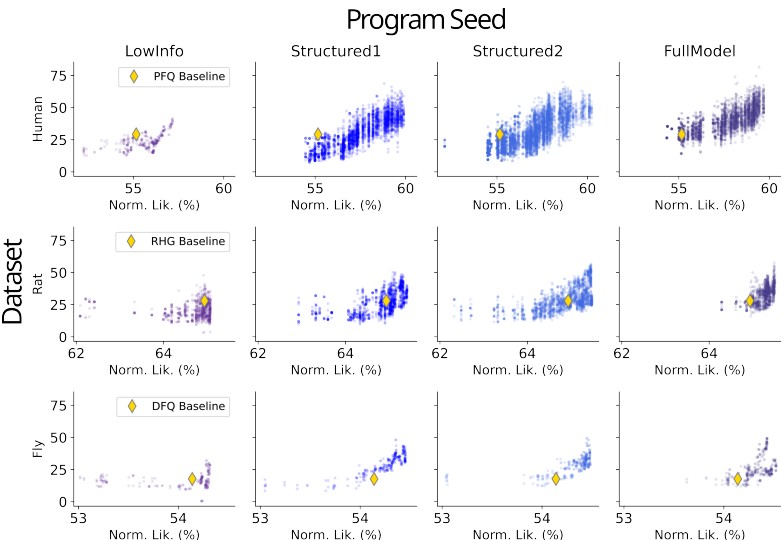

*Figure 12.* **Halstead Difficulty for generated programs given different seed programs.** Each row represents a different dataset (Human, Rat, Fly), each column a different seed program, and each point a different program.

We additionally explored a novel measure of complexity by prompting a large language model (LLM) to rate the relative complexity of our programs (see Appendix C.3.1 for the prompt used). Figure 13 plots the complexity score against their likelihood, for each of the generated programs from the best seed prompts for each dataset.

Here, to reduce LLM calls, we limit the Human programs to those with values above 56% normalized likelihood, which is why PFQ Baseline appears to be an outlier in Fig. 13 but not in 12. Far fewer programs were generated for Fruit Fly and Rat datasets so filtering was not required.

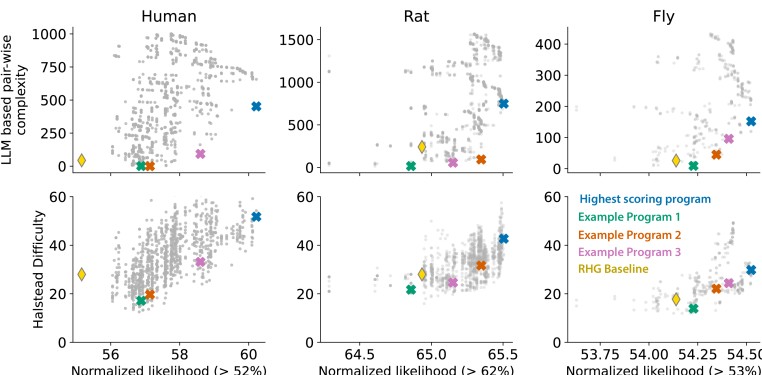

*Figure 13.* **LLM-based and Halstead Complexity Program Scores for Different Datasets.** Each column represents a different dataset (Human, Rat, Fly), and each point a different program.

### C.3.1. LLM-BASED PAIR-WISE COMPLEXITY SCORES

We use the prompt below to query an LLM to decide which of a pair of programs is less complex. These "preferences" are then used to compute the scores displayed in Figure 7 and Figure 13.

```
Below are 2 code snippets. Which of them is easier to understand?.
Code snippet 1:
<program 1>
Code snippet 2:
<program 2>
Please explain your reasoning. End your answer with "Final Answer: <ans>", where ans is 0 if the first,
snippet is simpler and/or easier to understand than the
second snippet. Otherwise, ans is 1. Do not output anything after that.
```

This comparison operation is applied to all programs in the dataset with MergeSort, which returns a ranked list from least to most complex. This rank is what is shown as LLM-based complexity score in Fig. 7 and Fig. 13.

### C.4. Quantifying variability in discovered models

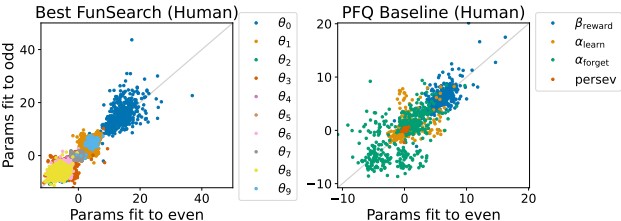

*Figure 14.* **Relationship between parameter values fit to even and odd splits of Human dataset.** Each color denotes a different parameter, and each dot a human subject in the dataset. The $x$-coordinate denotes the values of parameters fit to even sessions, the $y$-coordinate the values of parameters fit to odd. (left) Parameters for best CogFunSearch program, (right) parameters for PFQ Baseline.

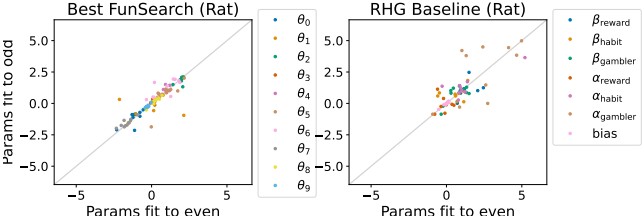

*Figure 15.* **Relationship between parameter values fit to even and odd splits of Rat dataset.** Each color denotes a different parameter, and each dot a rat in the dataset. The $x$-coordinate denotes the values of parameters fit to even sessions, the $y$-coordinate the values of parameters fit to odd. (left) Parameters for best CogFunSearch program, (right) parameters for RHG Baseline.

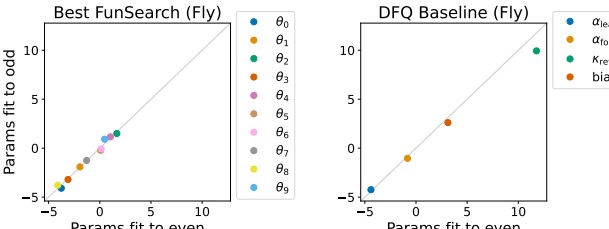

*Figure 16.* **Relationship between parameter values fit to even and odd splits of Fruit Fly dataset.** Each color/dot denotes a different parameter. Multiple dots are not shown because a single set of parameters was fit for all subjects. The $x$-coordinate denotes the values of parameters fit to even sessions, the $y$-coordinate the values of parameters fit to odd. (left) Parameters for best CogFunSearch program, (right) parameters for DFQ Baseline.

In order to assess the robustness of the fit parameters under the discovered program, we compare the values of different experimental parameters found across different data splits (e.g. even v. odd sessions). In Fig. 14, 15, and 16, we show the parameters recovered for the even/odd split for the best discovered program for each dataset. We see that parameters discovered across different splits of the data covary closely. We find these parameters to be even more consistent than those found under the cognitive model.

### C.5. Optimizability of the discovered models

A risk of our optimization pipeline is that we may evolve programs that are overadapted for the specific initial parameters used for optimization, and show extreme sensitivity to the random seed used for parameter fitting. In order to determine whether this was the case, we performed parameter fitting 50 times with different random seeds for both the best Rat program and the RHG cognitive model baseline, for comparison.

We explore this for the best Rat programs found with FullModel seed program, and find that the discovered programs do show a preference for the seed that was used during optimization (normalized likelihood on training seed = 67.206%, mean normalized likelihood across other seeds = $67.187 \pm 0.029$, $t = -4.447$, $p = 5 \times 10^{-5}$, only 15/49 seeds have larger values than seed=0) (Fig. 17). Regardless, the best Rat program continues to outperform RHG baseline across all seeds, and also appears to show less dependence on seed than the baseline (normalized likelihood $66.64 \pm 0.079$). Thus, while some specialization for the training seed is evident, we conclude that it does not seem to result in more brittle, hard-to-optimize programs.

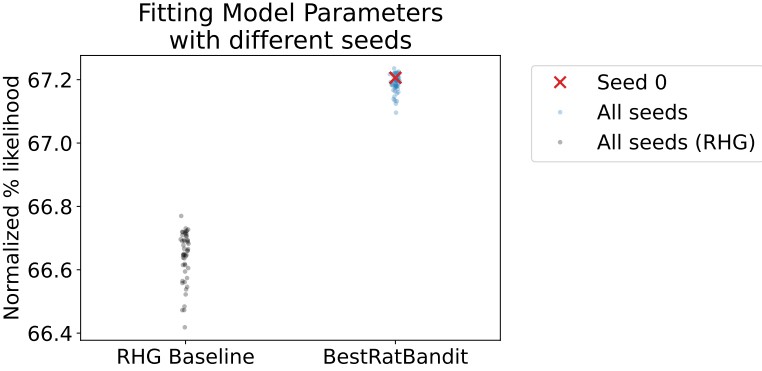

*Figure 17.* Evaluating whether evolved programs are overadapted to the seed used for optimization, when doing parameter fitting. We observe less dependence on seed than the baseline, and all random seeds continue to outperform all fits to RHG baseline.

Another risk of our pipeline is that we might learn programs that are overparameterized, with many different parameter values capable of yielding similar solutions. In this case, we would expect to be substantial variability in the parameters discovered when using different seeds. However, looking at the best CogFunSearch program, we find that the value of parameters discovered from different seeds tend to covary with that of the seed used for optimization. Shown in Fig. 18 are

the parameters discovered from different values.

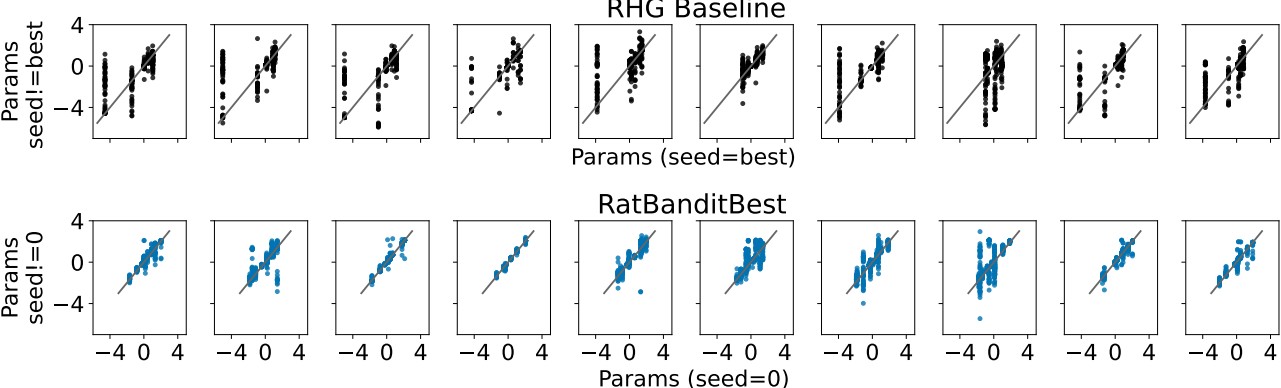

*Figure 18.* **Evaluating whether our models are overparameterized.** We plot the parameter values found by seed 0 against those found by other seeds, and find that they are generally covariant, suggesting that our models are not overparameterized.

## C.6. Rejection sampling for compute efficiency

We experimented with a form of rejection sampling for reducing the computational expense. Specifically, we maintain a set of scores $\boldsymbol{\Omega} = \{\Omega_0, \Omega_1, \ldots, \Omega_n\}$ for all programs evaluated thus far (see Section 3.2 for the details on how $\Omega$ is computed). We use the 90th quantile as a threshold $\tau$. When a new program $\phi_i$ is generated, we first evaluate it on a small subset of the subjects (in our experiments we used 20) to produce a proxy score $\Omega_i'$. If $\Omega_i' \geq \tau$, we compute $\Omega_i$ on the full set of subjects and add it to $\boldsymbol{\Omega}$; if not, we reject $\phi_i$.

This simple technique has the effect of only evolving promising programs. This myopic strategy naturally trades off exploration for computational savings, yet as Figure 19 and Table 2 demonstrate, the resulting programs significantly improve over the baseline (Wilcoxon signed rank tests on score improvement by subject: $W = 2203$ (human), 4 (rat), 1322 (flies); all with $p < 0.02$), and are competitive to the full runs.

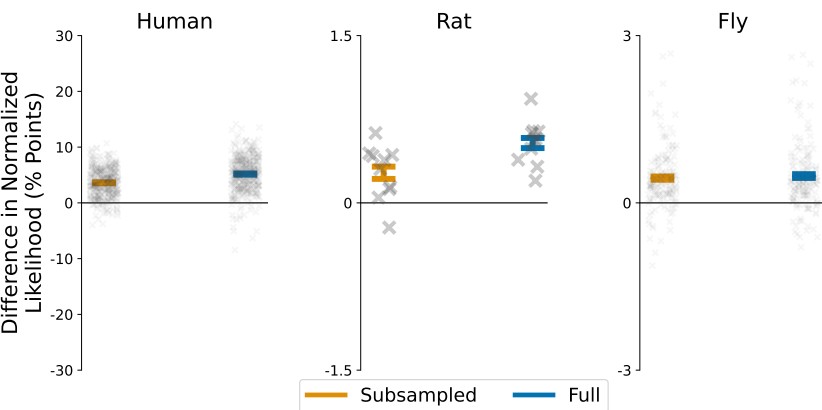

*Figure 19.* **Rejection sampling can produce accurate models at a fraction of the compute.** Despite trained with an order of magnitude fewer samples, the discovered models still significantly outperform the baseline ($p < 0.02$ for all datasets).

|  | Baseline | CogFunSearch (Subsampled) | CogFunSearch (Full) |
|---|---|---|---|
| **Human** | $55.77 \pm 0.64$ | $59.38 \pm 0.62$ | $60.93 \pm 0.63$ |
| **Rat** | $66.63 \pm 0.53$ | $67.03 \pm 1.27$ | $67.22 \pm 1.25$ |
| **Fly** | $53.99 \pm 0.24$ | $54.58 \pm 0.56$ | $54.62 \pm 0.57$ |

*Table 2.* Normalized likelihood scores (as percentages) and standard error, as computed over held-out test subjects, for the evaluated methods on each of the datasets.

## D. Model Parameters

In this section we present the numerical values and distributions for the parameters in both the baseline and discovered models, for each dataset.

### D.1. Human parameters

Figure 20 displays the distribution of parameter values of the best model discovered by CogFunSearch for the human dataset, after fitting the parameters to the data. Table 3 displays the average and standard errors for these parameters.

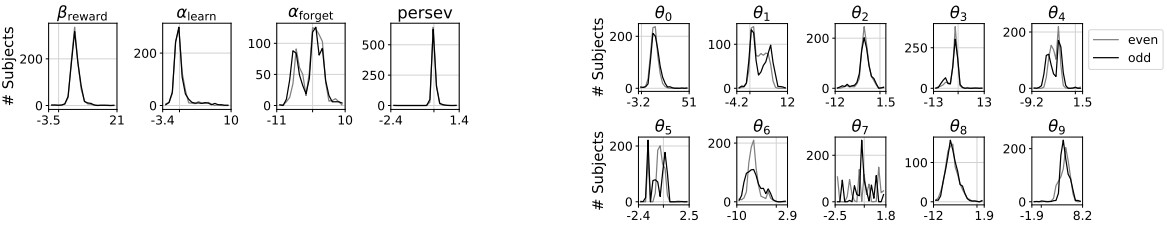

*Figure 20.* Distribution of parameter values of the best model discovered by CogFunSearch for the human dataset.

| PFQ Baseline | | BestHumanBandit | | | |
|---|---|---|---|---|---|
| **Parameter** | **Value** | **Parameter** | **Value** | **Assigned name** | **Transformed Value** |
| $\beta_{\text{reward}}$ | $5.9 \pm 1.7$ | $\theta_0$ | $15 \pm 4.5$ | `beta_r` | $14 \pm 3.7$ |
| $\alpha_{\text{learn}}$ | $0.19 \pm 1.6$ | $\theta_1$ | $3.1 \pm 2.6$ | `lapse` | $0.83 \pm 0.2$ |
| $\alpha_{\text{forget}}$ | $-0.6 \pm 3.7$ | $\theta_2$ | $-4.1 \pm 1.6$ | `prior` | $0.044 \pm 0.091$ |
| persev | $0.0018 \pm 0.12$ | $\theta_3$ | $-2.3 \pm 2.5$ | `alpha_exploration_rate` | $0.2 \pm 0.18$ |
| | | $\theta_4$ | $-4.4 \pm 1.2$ | `decay_rate` | $0.026 \pm 0.05$ |
| | | $\theta_5$ | $-0.5 \pm 0.65$ | `attention_bias1` | $-0.5 \pm 0.65$ |
| | | $\theta_6$ | $-5.7 \pm 1.8$ | `attention_bias2` | $-5.7 \pm 1.8$ |
| | | $\theta_7$ | $-0.14 \pm 1.1$ | `perseveration_strength` | $0.75 \pm 0.48$ |
| | | $\theta_8$ | $-6.6 \pm 1.7$ | `switch_strength` | $-6.6 \pm 1.7$ |
| | | $\theta_9$ | $4.6 \pm 0.96$ | `lambda_param` | $1 \pm 0.036$ |
| | | | | `gamma` | $4.6 \pm 0.92$ |
| | | | | `temperature` | $4.6 \pm 0.92$ |
| | | | | `beta_p` | $4.6 \pm 0.92$ |

*Table 3.* Parameter values for PFQ Baseline (left) and HumanBanditBest discovered FunSearch program. For the discovered program, we also show the values after they are transformed inside the FunSearch program, as well as the names they are assigned by the programs. Shown are the means and standard deviations across all subjects and across both even and odd session splits. Note that $\theta_9$ additional named variables transform $\theta_9$ – they typically reflect attempts to index further into the parameter array than the length (which in Jax, defaults to selecting the last element).

## D.2. Rat parameters

Figure 21 displays the distribution of parameter values of the best model discovered by CogFunSearch for the rat dataset, after fitting the parameters to the data. Table 4 displays the average and standard errors for these parameters.

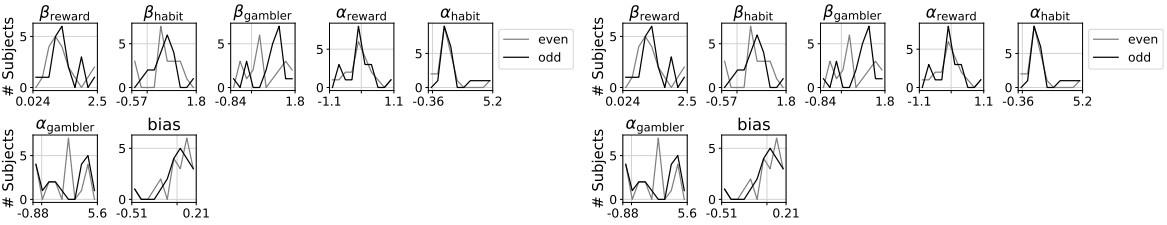

*Figure 21.* Distribution of parameter values of the best model discovered by CogFunSearch for the rat dataset.

| RHG Baseline | | Best Rat Program | | | |
|---|---|---|---|---|---|
| **Parameter** | **Value** | **Parameter** | **Value** | **Assigned name** | **Transformed Value** |
| $\beta_{\text{reward}}$ | $1.2 \pm 0.54$ | $\theta_0$ | $-1.1 \pm 0.46$ | `alpha_r` | $0.35 \pm 0.074$ |
| $\beta_{\text{habit}}$ | $0.63 \pm 0.48$ | $\theta_1$ | $-0.41 \pm 1.4$ | `alpha_h` | $0.55 \pm 0.21$ |
| $\beta_{\text{gambler}}$ | $0.64 \pm 0.66$ | $\theta_2$ | $2 \pm 0.2$ | `alpha_g` | $0.98 \pm 0.025$ |
| $\alpha_{\text{reward}}$ | $-0.084 \pm 0.45$ | $\theta_3$ | $-0.031 \pm 0.48$ | `alpha_q` | $0.6 \pm 0.1$ |
| $\alpha_{\text{habit}}$ | $1.5 \pm 1.2$ | $\theta_4$ | $0.72 \pm 0.49$ | `beta_r` | $1.1 \pm 0.32$ |
| $\alpha_{\text{gambler}}$ | $2.3 \pm 2$ | $\theta_5$ | $0.76 \pm 0.87$ | `beta_h` | $1.2 \pm 0.46$ |
| bias | $0.0059 \pm 0.16$ | $\theta_6$ | $1.1 \pm 0.74$ | `beta_g` | $1.5 \pm 0.56$ |
| | | $\theta_7$ | $-1.7 \pm 0.33$ | `beta_q` | $0.18 \pm 0.056$ |
| | | $\theta_8$ | $0.27 \pm 0.25$ | `bias` | $0.27 \pm 0.25$ |
| | | $\theta_9$ | $-0.26 \pm 0.12$ | `beta_q_bias` | $-0.26 \pm 0.12$ |
| | | | | `beta_h_bias` | $-0.26 \pm 0.12$ |
| | | | | `exploration_noise` | $0.57 \pm 0.051$ |
| | | | | `lapse_rate` | $0.43 \pm 0.028$ |
| | | | | `gamma_q` | $0.57 \pm 0.051$ |
| | | | | `gamma_w` | $0.64 \pm 0.02$ |
| | | | | `alpha_bias` | $0.43 \pm 0.028$ |
| | | | | `reward_bias` | $-0.26 \pm 0.12$ |
| | | | | `bias_1` | $-0.26 \pm 0.12$ |

*Table 4.* Parameter values for RHG Baseline (left) and Best Rat Program discovered by CogFunSearch. For the discovered program, we also show the values after they are transformed inside the CogFunSearch program, as well as the names they are assigned by the programs. Shown are the means and standard deviations across all subjects and across both even and odd session splits. Note that $\theta_9$ additional named variables transform $\theta_9$, and typically reflect attempts to index further into the parameter array than the length (which in Jax, defaults to selecting the last element).

## D.3. Fly

Table 5 displays the average and standard errors for the parameters of the best model discovered by CogFunSearch for the fly dataset, after fitting the parameters to the data.

| DFQ Baseline | | BestFlyBandit | | | |
|---|---|---|---|---|---|
| **Parameter** | **Value** | **Parameter** | **Value** | **Assigned name** | **Transformed Value** |
| $\alpha_{\text{learn}}$ | -3.8, -4.1 | $\theta_0$ | -3.8, -4.1 | `params[0]/10` | -0.38,-0.41 |
| $\alpha_{\text{forget}}$ | -1.9, -1.9 | $\theta_1$ | -1.9, -1.9 | `lr` | 0.13,0.14 |
| $\kappa_{\text{reward}}$ | 1.7, 1.5 | $\theta_2$ | 1.7, 1.5 | `temperature` | 5.3,4.5 |
| bias | -3.1, -3.2 | $\theta_3$ | -3.1, -3.2 | `lapse_rate` | 0.043,0.039 |
| | | $\theta_4$ | 1, 1.2 | `alpha` | 2.8,3.2 |
| | | $\theta_5$ | 0.083, -0.22 | `bias[0]` | 0.74,0.59 |
| | | $\theta_6$ | 0.14, -0.071 | `bias[1]` | 0.76,0.66 |
| | | $\theta_7$ | -1.3, -1.3 | `forget_rate` | 0.24,0.25 |
| | | $\theta_8$ | -4.1, -3.8 | `decay_factor` | 0.98,0.98 |
| | | $\theta_9$ | 0.47, 0.91 | `exploitation_rate` | 0.61,0.71 |

*Table 5.* Parameter values for DFQ Baseline (left) and FlyBanditBest discovered FunSearch program. For the discovered program, we also show the values after they are transformed inside the FunSearch program, as well as the names they are assigned by the programs. The two values shown correspond to the parameter values for (even, odd) subjects. Since only one set of parameters is recovered per dataset split, no error is provided.

# E. Baseline Programs

## E.1. Human Cognitive Model Baseline

We used the following implementation of the PFQ Baseline from Eckstein et al. (2024).

```python
def agent(
    params: chex.Array,
    choice: int,
    reward: int,
    agent_state: Optional[chex.Array],
) -> Tuple[chex.Array, chex.Array]:
  """Cognitive model describing human behavior on a multi-armed bandit task.

  Assumes the agent is presented with four options on each trial.

  Args:
    params: a list containing [beta_r, alpha_learn, alpha_forget, and p]
    choice: Choice made by the agent on the previous trial. 0, 1, 2, or 3
    reward: Reward received by the agent on the previous trial. A scalar between
      0 and 100.
    agent_state: [Q1, Q2, Q3, Q4, previous_choice] If None, assumes this is the
      first trial of a new session and defaults to [0, 0, 0, 0, -1].

  Returns:
    choice_logits: The probabilities that the agent will choose option 0, 1, 2,
      or 3 on the next trial, expressed as logits.
    agent_state: New agent state
  """

  if agent_state is None:
    agent_state = jnp.array(([0, 0, 0, 0, -1]))

  qs = agent_state[:4]
  prev_choice = jnp.int32(agent_state[5])

  beta_r = params[0]
  alpha_learn = 1 / (1 + jnp.exp(-params[1]))
  alpha_forget = 1 / (1 + jnp.exp(-params[2]))
  p = params[3]

  # One-back perseveration param should be 0 on the first trial of the session
  # We indicate this using prev_choice of -1
  p = p * (prev_choice != -1)

  # Update Q for chosen action
  choice = jnp.int32(choice)
  qs = qs.at[choice].set(alpha_learn * (reward - qs[choice]) + qs[choice])

  # Update Q for unchosen actions using a mask
  mask = jnp.ones_like(qs, dtype=bool)
  mask = mask.at[choice].set(False)
  qs = jnp.where(mask, qs * alpha_forget, qs)

  # Values for choice: Qs plus bonus for p
  qs_for_choice = qs.at[prev_choice].set(p + qs[prev_choice])

  agent_state = jnp.append(qs, choice)

  # Compute choice logits
  choice_logits = beta_r * qs_for_choice

  return choice_logits, agent_state
```

### E.2. Rat Cognitive Model Baseline

We used the following implementation of the RHG Baseline model from Miller et al. (2021).

```python
def agent(
    params: chex.Array,
    choice: int,
    reward: int,
    agent_state: Optional[chex.Array],
) -> Tuple[chex.Array, chex.Array]:
    """Cognitive model describing rat behavior on a binary two-armed bandit task.

    Args:
      params: Model params. Different parameters are used for different rats.
      choice: Previous choice. Values: 0 or 1.
      reward: Previous reward. Values: 0 or 1.
      agent_state: Previous state of the agent

    Returns:
      choice_logits: Vector of shape (2,) with the probabilities that the rat will
        choose option 0 or 1 on the next trial, expressed as logits.
      agent_state: New state of the agent, after observing the previous choice and
        reward.
    """
    if agent_state is None:
        agent_state = jnp.zeros((3,))

    beta_r = params[0]
    beta_h = params[1]
    beta_g = params[2]
    alpha_r = 1 / (1 + jnp.exp(-params[3]))
    alpha_h = 1 / (1 + jnp.exp(-params[4]))
    alpha_g = 1 / (1 + jnp.exp(-params[5]))
    bias = params[6]

    # Convert choice and reward 0/1 to +-1
    choice_for_update = 2*choice-1
    reward_for_update = 2*reward-1

    # Update R
    agent_state = agent_state.at[0].set(
        alpha_r * agent_state[0] +
        (1 - alpha_r) * reward_for_update * choice_for_update
    )
    # Update H
    agent_state = agent_state.at[1].set(
        alpha_h * agent_state[1] + (1 - alpha_h) * choice_for_update
    )
    # Update G
    agent_state = agent_state.at[2].set(
        alpha_g * agent_state[2]
        + (1 - alpha_g)
        * (choice_for_update - reward_for_update * choice_for_update)
    )

    choice_term = 0

    # Update choice term from R.
    choice_term += beta_r * agent_state[0]

    # Update choice term from H.
    choice_term += beta_h * agent_state[1]

    # Update choice term from G.
    choice_term += beta_g * agent_state[2]

    # Update choice term from bias.
    choice_term += bias

    choice_logits = jnp.array([-1, 1]) * choice_term

    return choice_logits, agent_state
```

### E.3. Fly Cognitive Model Baseline

We use the following implementation of the DFQ Baseline model from Ito & Doya (2009). See Ito & Doya (2009) and Mohanta (2022) for more details.

```python
def agent(
    params: chex.Array,
    choice: int,
    reward: int,
    agent_state: Optional[chex.Array]
) -> Tuple[chex.Array, chex.Array]:
  """Cognitive model describing fly behavior on a binary two-armed bandit task.

  Args:
    params: Fit parameters of the model.
    choice: The choice made by a fly on this trial. 0 or 1
    reward: The reward received by a fly on this trial. 0 or 1
    agent_state: The current state of the cognitive model.

  Returns:
    choice_logits: The probabilities that the fly will choose option 0 or 1 on
      the next trial, expressed as logits.
    agent_state: The updated state of the cognitive model.
  """
  # Initialize Q-values to 0.5.
  if agent_state is None:
    agent_state = jnp.array((0.5, 0.5))

  # Unpack and rename parameters for readability.
  alpha_learn_logit = params[0]
  alpha_forget_logit = params[1]
  kappa_reward = params[2]
  kappa_omission = params[3]
  alpha_learn = 1 / (1 + jnp.exp(-alpha_learn_logit))
  alpha_forget = 1 / (1 + jnp.exp(-alpha_forget_logit))
  qs = agent_state

  # Chosen action's Q-value will be updated towards this target value.
  learn_target = kappa_reward * reward + kappa_omission * (1 - reward)

  # Update Q-value for chosen action.
  qs = qs.at[choice].set(
      alpha_learn * (learn_target - qs[choice]) + (1 - alpha_learn) * qs[choice]
  )
  # Update Q-value for unchosen action: decay towards 0.
  qs = qs.at[1 - choice].set((1 - alpha_forget) * (qs[1 - choice]))

  logits = qs
  new_agent_state = qs
  return logits, new_agent_state
```

# F. Seed programs

In this section we list the seed programs provided to CogFunSearch for the rat dataset. Note that the programs used for human and fly datasets are functionally identical, with the exception of the FullModel seed program.

## F.1. LowInfo seed program

```python
"""Fits a model to a dataset."""

def score_function(data_id):
  """Fits parameters of a model and returns the normalized likelihood.

  Args:
    data_id: Data id.

  Returns:
    Normalized likelihood of the fit model on held out data, in range [0, 1].
  """
  dataset = load_dataset(data_id)
  normalized_likelihood = twofold_crossvalidation(dataset, model)
  return normalized_likelihood

def model(params, x, y, state):
  """Implements a predictive model.

  Args:
    params: parameters of the model
    x: first input
    y: second input
    state: current state of the model

  Returns:
    output: output of the model
    state: updated state of the model
  """
  if state is None:
    state = jnp.zeros((10,))

  # Fill in code here

  output = jnp.array([0, 0])

  return output, state
```

## F.2. Structured1 seed program

```python
"""Finds a likelihood function for a binary bandit task for the rat dataset."""

def score_function(rat_id):
  """Computes normalized likelihood of a dataset given a model.

  Retrieves saved datasets from disk. Datasets are stored as JSON files with
  keys "choices" and "rewards". Each of these is a list of lists of integers,
  representing the choices and rewards made by the rat on each session.

  Computes twofold cross-validation, using even-numbered and odd-numbered
  sessions.

  Args:
    rat_id: Rat id.

  Returns:
    Likelihood of the fit model, in range [0, 1].
  """
  dataset = load_rat_data(rat_i=rat_id)

  normalized_likelihood = twofold_crossvalidation(dataset, agent)

  return normalized_likelihood
```

```python
def agent(params, choice, reward, agent_state):
  """Cognitive model describing rat behavior on a binary two-armed bandit task.

  Args:
    params: Model params. Different parameters are used for different rats.
    choice: Previous choice. Values: 0 or 1.
    reward: Previous reward. Values: 0 or 1.
    agent_state: Previous state of the agent

  Returns:
    choice_logits: Vector of shape (2,) with the probabilities that the rat will
      choose option 0 or 1 on the next trial, expressed as logits.
    agent_state: New state of the agent, after observing the previous choice and
      reward.
  """
  if agent_state is None:
    agent_state = jnp.zeros((3,))

  param0 = params[0]
  param1 = 1 / (1 + jnp.exp(-params[1]))

  # Fill in code here

  choice_logits = jnp.array([0, 0])

  return choice_logits, agent_state
```

## F.3. Structured2 seed program

For conciseness, we use `...` to indicate a section of the prompt that is the same as in Structured1.

```python
...
def agent(params, choice, reward, agent_state):
  """
  ...
  """
  # Do not remove comments or TODOs from this program.

  # TODO: Initialize agent state.
  if agent_state is None:
    agent_state = jnp.zeros((3,))

  # TODO: Define all parameters.
  # Each parameter assignment should be a single line.
  # Change the names of the parameters to be descriptive.
  # Include comments describing the parameter transformation, if applicable.
  param0 = params[0]  # Placeholder parameter definition.
  param1 = 1 / (1 + jnp.exp(-params[1]))  # Placeholder parameter definition with logistic function.

  # TODO: Update agent_state.
  # Do not update more than one element at a time.
  # Each component of the agent state update should be a single line.
  # Include comments describing the computation in each line.
  #   agent_state[i] = ...

  # TODO: Compute choice_logits.
  # Each component of the choice logit computation should be a single line.
  # Include comments describing the computation in each line.
  choice_logits = jnp.array([0, 0])

  return choice_logits, agent_state
```

## F.4. FullModel seed program

For conciseness, we use `...` to indicate a section of the prompt that is the same as in Structured1.

```python
...
def agent(params, choice, reward, agent_state):
  """
```

```python
    ...
    """
    if agent_state is None:
      agent_state = jnp.zeros((3,))

    beta_r = params[0]
    beta_h = params[1]
    beta_g = params[2]
    alpha_r = 1 / (1 + jnp.exp(-params[3]))
    alpha_h = 1 / (1 + jnp.exp(-params[4]))
    alpha_g = 1 / (1 + jnp.exp(-params[5]))
    bias = params[6]

    # Convert choice and reward 0/1 to +-1
    choice_for_update = 2*choice-1
    reward_for_update = 2*reward-1

    # Update R
    agent_state = agent_state.at[0].set(
        alpha_r * agent_state[0] +
        (1 - alpha_r) * reward_for_update * choice_for_update
    )
    # Update H
    agent_state = agent_state.at[1].set(
        alpha_h * agent_state[1] + (1 - alpha_h) * choice_for_update
    )
    # Update G
    agent_state = agent_state.at[2].set(
        alpha_g * agent_state[2]
        + (1 - alpha_g)
        * (choice_for_update - reward_for_update * choice_for_update)
    )

    choice_term = 0

    # Update choice term from R.
    choice_term += beta_r * agent_state[0]

    # Update choice term from H.
    choice_term += beta_h * agent_state[1]

    # Update choice term from G.
    choice_term += beta_g * agent_state[2]

    # Update choice term from bias.
    choice_term += bias

    choice_logits = jnp.array([-1, 1]) * choice_term

    return choice_logits, agent_state
```

# G. Best Discovered Programs

Here we show the best discovered program for each dataset. Quantitative results from these programs are featured in Fig. 1 (see Sec. 4.1). We note that different prompts elicited different best programs depending on the dataset: Best Human Program was produced with Structured2 prompt, whereas Best Rat Program and Best Fly Program were produced with FullModel prompt.

## G.1. Human Structured2

This program was the overall best discovered program for Human. It was generated using the Structured2 seed program and occurred after $1, 239, 338$ samples.

```python
def agent(params, choice, reward, agent_state = None):
  num_params = 13

  params = jnp.clip(params, -5, 5)

  beta_r = jnp.clip(jax.nn.softplus(params[0]), 0.01, 20)
  lapse = jnp.clip(jax.nn.sigmoid(params[1]), 0.01, 0.99)
  prior = jnp.clip(jax.nn.softplus(params[2]), 0.01, 0.99)
  alpha_exploration_rate = jnp.clip(jax.nn.sigmoid(params[3]), 0.01, 0.99)
  decay_rate = jnp.clip(jax.nn.sigmoid(params[4]), 0.01, 0.99)
  attention_bias1 = params[5]
  attention_bias2 = params[6]
  perseveration_strength = jax.nn.softplus(params[7])
  switch_strength = params[8]
  lambda_param = jnp.clip(jax.nn.softplus(params[9]), 0.0, 1.0)
  gamma = jax.nn.softplus(params[10]) # Loss aversion parameter
  temperature = jnp.clip(jax.nn.softplus(params[11]) + 1e-6, 1e-6, 100) #Softmax temperature
  beta_p = jax.nn.softplus(params[12])

  if agent_state is None:
    q_values = jnp.ones((4,)) * prior
    old_choice = -1
    trial_since_last_switch = 0
    exploration_rate = alpha_exploration_rate
    cumchoice = jnp.zeros((4,))
  else:
    q_values = agent_state[:4]
    old_choice = agent_state[4]
    trial_since_last_switch = agent_state[5]
    exploration_rate = agent_state[6]
    cumchoice = agent_state[7:11]

  if choice is not None and reward is not None:
    delta = reward - gamma*(1-reward) - q_values[choice]
    q_values = q_values.at[choice].set(q_values[choice] + delta)

    trial_since_last_switch = jnp.where(choice == old_choice, trial_since_last_switch + 1, 0)
    exploration_rate = exploration_rate * (1 - 1e-3) # decay exploration rate slowly
    cumchoice = cumchoice.at[choice].set(cumchoice[choice] + 1)

  q_values = (1 - exploration_rate) * q_values + exploration_rate * jnp.mean(q_values)
  q_values = q_values * decay_rate

  choice_probs = (1 - lapse) * jax.nn.softmax(beta_r * q_values / temperature + beta_p * jnp.log(
      1 + cumchoice)) + lapse / 4
  choice_logits = jnp.log(choice_probs)

  if choice is not None:
    perseveration_bonus = (choice == old_choice) * perseveration_strength * jax.nn.one_hot(
        choice, num_classes=4)
    switch_bonus = (choice != old_choice) * switch_strength * jax.nn.one_hot(choice, num_classes=4)
    attention_bonus1 = attention_bias1 * jax.nn.one_hot(old_choice, num_classes=4)
    attention_bonus2 = attention_bias2 * jax.nn.one_hot((choice + 2) % 4, num_classes=4)

    choice_logits = (
        choice_logits + perseveration_bonus + switch_bonus + attention_bonus1 + attention_bonus2 +
        jax.nn.one_hot(choice, 4) * jnp.log(trial_since_last_switch + 1))

  agent_state = jnp.concatenate(
      [q_values, jnp.array([choice, trial_since_last_switch, exploration_rate]),
      cumchoice])

  return choice_logits, agent_state
```

## G.2. Rat FullModel

This program was the overall best discovered program for Rat. It was generated using the FullModel seed program and occurred after 292, 690 samples.

```python
def agent(params, choice, reward, agent_state = None):
  if agent_state is None:
    agent_state = jnp.zeros(18)
  alpha_r = jax.nn.sigmoid(params[0])
  alpha_h = jax.nn.sigmoid(params[1])
  alpha_g = jax.nn.sigmoid(params[2])
  alpha_q = jax.nn.sigmoid(params[3])
  beta_r = jax.nn.softplus(params[4])
  beta_h = jax.nn.softplus(params[5])
  beta_g = jax.nn.softplus(params[6])
  beta_q = jax.nn.softplus(params[7])
  bias = params[8]
  beta_q_bias = params[9]
  beta_h_bias = params[10]
  exploration_noise = jax.nn.softplus(params[11])
  lapse_rate = jax.nn.sigmoid(params[12])
  gamma_q = jax.nn.softplus(params[13])
  gamma_w = 1./(1+jax.nn.softplus(params[14]))
  alpha_bias = jax.nn.sigmoid(params[15])
  reward_bias = params[16]
  bias_1 = params[17]

  alpha_r = alpha_r + 0.1
  alpha_h = alpha_h + 0.1
  alpha_g = alpha_g + 0.1
  alpha_q = alpha_q + 0.1

  choice_for_update = 2 * choice - 1
  reward_for_update = 2 * reward - 1 + reward_bias

  agent_state = agent_state.at[0].set(
      alpha_r * agent_state[0] + (1 - alpha_r) * reward_for_update * choice_for_update
  )
  agent_state = agent_state.at[1].set(
      alpha_h * agent_state[1] + (1 - alpha_h) * choice_for_update
  )
  agent_state = agent_state.at[2].set(
      alpha_g * agent_state[2]
      + (1 - alpha_g) * choice_for_update * (1 - reward_for_update)
  )

  state_q = agent_state[3:5]
  state_w = agent_state[5:7]

  state_q = state_q.at[choice].set(alpha_q * state_q[choice] + (
      (1 - alpha_q) * reward_for_update +
      alpha_bias * gamma_q * state_w[1-choice]))
  state_w = state_w.at[choice].set(alpha_q * state_w[choice] + (
      (1 - alpha_q) * reward_for_update +
      alpha_bias * gamma_w * state_w[1-choice]))

  agent_state = agent_state.at[3:5].set(state_q)
  agent_state = agent_state.at[5:7].set(state_w)

  Q = agent_state[3:5]
  W = agent_state[5:7]
  choice_term = (
      beta_r * agent_state[0]
      + beta_h * agent_state[1]
      + beta_g * agent_state[2]
      + beta_q * (W[0] * Q[0] - W[1] * Q[1])
      + bias
      + beta_q_bias * (Q[0] - Q[1])
      + beta_h_bias * agent_state[1] + bias_1
  )
  choice_logits = (
      lapse_rate * jnp.log(jnp.array([0.5, 0.5])) +
      (1 - lapse_rate) * jnp.array([-1, 1]) * choice_term)
  return choice_logits, agent_state
```

### G.3. Fruit Fly FullModel

This program was the overall best discovered program for FlyBandit. It was generated using the FullModel seed program and occurred after $195,962$ samples.

```python
def agent(params, choice, reward, agent_state = None):
  n_params_v2 = 10
  if agent_state is None:
    agent_state = jnp.repeat(params[0] / n_params_v2, 2)
  else:
    agent_state = agent_state

  lr = jax.nn.softplus(params[1])
  temperature = jnp.exp(params[2])
  lapse_rate = jax.nn.sigmoid(params[3])
  alpha = jnp.exp(params[4])  # alpha > 0
  bias = jax.nn.softplus(params[5:7])  # Bias should be positive
  forget_rate = jax.nn.softplus(params[7])
  decay_rate = jax.nn.softplus(params[8])
  exploitation_rate = jax.nn.sigmoid(params[9])

  q_vals = agent_state
  rpe = reward - q_vals[choice]
  p_choice = jax.nn.sigmoid(bias[1] * (q_vals[1 - choice] - q_vals[choice]))

  decay_factor = jnp.exp(-decay_rate)
  delta_qval = lr * rpe * p_choice
  updated_q_values_choice = q_vals.at[choice].set(q_vals[choice] + delta_qval)

  updated_q_values = jax.lax.select(
      reward, updated_q_values_choice, jnp.copy(q_vals) * decay_factor)
  updated_q_values = updated_q_values.at[1 - choice].set(
      updated_q_values[1 - choice]
      - forget_rate * jnp.maximum(0, rpe * (1 - p_choice) * decay_factor)
  )  # Changed this line to reduce overwriting

  updated_q_values = jnp.clip(updated_q_values, -alpha, alpha)
  updated_q_values = updated_q_values * (1 - lapse_rate)

  choice_logits = (
      exploitation_rate * jnp.exp(bias + temperature * updated_q_values)
      + (1 - exploitation_rate) * jnp.exp(bias)
      + 1e-5
  )

  return choice_logits, updated_q_values
```

## H. Seed program comparison

Here we show example best CogFunSearch programs that are discovered for the Rat dataset under different seed programs (Sec. F). We note that these programs differ from the best found programs, as they use a later cutoff. This earlier cutoff was used here to permit comparison with programs from other prompts given a similar number of samples.

### H.1. Rat LowInfo

Best LowInfo program given a cutoff of 165,000 samples.

```python
def model(params, x, y, state = None):
  nparams = len(params)
  z = jnp.max(jnp.array([jnp.abs(x * y) * 0.5, jnp.tanh(x * y) * 0.5]))
  z1 = jnp.sqrt(jnp.abs(x))
  z2 = jnp.sqrt(jnp.abs(y))
  x_clip = jnp.clip(x, -20, 20)
  y_clip = jnp.clip(y, -10, 10)
  z_clip = jnp.clip(z, -20, 20)

  if state is None:
    state = jnp.zeros((3,))

  state_mean = jnp.mean(state)
  x1 = (params[0] * z_clip + params[1] * x_clip + params[2] * y_clip +
```

```
        params[3] * state_mean + params[4])
x2 = (params[5] * x_clip + params[6] * jnp.sin(y_clip) + params[7] * z_clip +
        params[8] * state[1] + params[9])
output = jnp.array([x1, x2])
alpha = 0.2
beta = 0.7
gamma = 0.8
state = jnp.array([
    gamma * (x1 + 1) + (1 - gamma) * state[0] + params[10] * x_clip * x_clip +
    params[11] * y_clip * y_clip,
    beta * (x2 + 1) + (1 - beta) * state[1] + params[12] * z_clip +
    params[13] * z1,
    alpha * (
    params[14] * x_clip + params[15] * y_clip + params[16] * state[2]) * (
    1 + jnp.abs(x_clip) + jnp.abs(y_clip)) - (
    alpha * (state[2] + 2) + params[17] * x_clip * y_clip)
])
return output, state
```

## H.2. RatBandit Structured1

Best Structured1 program given a cutoff of 165,000 samples.

```
def agent(params, choice, reward, agent_state = None):
  num_params = 16
  alpha = jax.nn.softplus(params[0])
  beta = jax.nn.softplus(params[1])
  decay = jax.nn.sigmoid(params[2])
  initial_bias = jax.nn.softplus(params[3])
  bias_decay = jax.nn.softplus(params[4])
  exploration_noise = jax.nn.softplus(params[5])
  alpha_pe_pos = jax.nn.softplus(params[6])
  alpha_pe_neg = jax.nn.softplus(params[7])
  decay_pe = jax.nn.softplus(params[8])
  temperature = jax.nn.softplus(params[9]) + 0.1
  lapse_rate = jax.nn.sigmoid(params[10])
  initial_q_value = jax.nn.softplus(params[11])
  choice_bias = params[12]
  exploration_ratio = jax.nn.softplus(params[13])
  initial_bias_direction = params[14]
  beta_pe = jax.nn.softplus(params[15])

  if agent_state is None:
    initial_Q = jnp.full(2, initial_q_value)
    initial_bias = jax.nn.sigmoid(initial_bias * jnp.array(
            [initial_bias_direction, 1 - initial_bias_direction])) * 5
    initial_pe = jnp.zeros(2)
    agent_state = {
        'Q_values': initial_Q,
        'bias': initial_bias,
        'PE': initial_pe,
    }

  delta = reward - agent_state['Q_values'][choice]

  Q_values = agent_state['Q_values'].at[choice].set(
        agent_state['Q_values'][choice] + alpha * delta)
  Q_values = (1 - decay) * Q_values + decay * initial_q_value

  bias = (1 - bias_decay) * agent_state['bias']
  bias += initial_bias * jnp.array([1 - choice, choice])

  pe_pos = jnp.maximum(delta, 0)
  pe_neg = -jnp.minimum(delta, 0)
  pe = (1 - decay_pe) * agent_state['PE']
  pe += alpha_pe_pos * pe_pos * jnp.array([1 - choice, choice])
  pe -= alpha_pe_neg * pe_neg * jnp.array([1 - choice, choice])

  exploration_noise = exploration_noise * jax.random.beta(
        jax.random.PRNGKey(0), 1, exploration_ratio, shape=(2,))
  exploration_noise = exploration_noise - jnp.mean(exploration_noise)

  choice_logits = beta * (Q_values + bias
        ) / temperature + exploration_noise + pe + choice_bias * jnp.array(
                [1, -1]) + beta_pe * pe
  choice_logits = jnp.clip(choice_logits, -10, 10)
```

```
    #Add Lapse Rate
    lapse_prob = lapse_rate /2
    choice_logits = choice_logits + lapse_prob + lapse_prob

    agent_state = {
        'Q_values': Q_values,
        'bias': bias,
        'PE': pe,
    }
    return choice_logits, agent_state
```

## H.3. Rat Structured2

Best Structured2 program given a cutoff of 165,000 samples.

```
def agent(params, choice, reward, agent_state = None):

    alpha = jax.nn.softplus(params[0:2])
    inv_temp = jax.nn.softplus(params[2])
    lapse_rate = jax.nn.softplus(params[3])   # Changed sigmoid to softplus
    stickiness = jax.nn.softplus(params[4])
    decay_rate = jax.nn.softplus(params[5])   # Changed sigmoid to softplus
    omega = jax.nn.softplus(params[6:8])   # two omega params
    bias = params[8]
    init_q = params[9:11]
    sigma = jax.nn.softplus(params[11])
    exploration_rate = jax.nn.softplus(params[12])
    momentum = jax.nn.softplus(params[13])
    omega_factor = jax.nn.softplus(params[14])
    lapse_mode = jax.nn.softplus(params[15])
    lapse_alpha = jax.nn.softplus(params[16])

    if agent_state is None:
      agent_state = jnp.concatenate(
              [init_q, jnp.zeros(2) + 0.5,
               jnp.array([0.5]), jnp.array([0])])

    q_values = agent_state[:2]
    stickiness_state = agent_state[2:4]
    lapse_rate_state = agent_state[4]
    trial_count = agent_state[5]

    reward_noisy = reward + lapse_mode * jax.random.normal(
          jax.random.PRNGKey(0), shape=()) * sigma
    reward_noisy = jnp.clip(reward_noisy, 0, 1)
    delta = reward_noisy - q_values[choice]

    q_values_updated = jnp.where(
        choice == 0,
        jnp.array(
              [q_values[0] + alpha[0] * delta,
               q_values[1] - omega[0] * alpha[0] * delta * omega_factor]),
        jnp.array(
              [q_values[0] - omega[1] * alpha[1] * delta * omega_factor,
               q_values[1] + alpha[1] * delta]),
    )

    # Updated stickiness calculation
    stickiness_state = (
          (1 - decay_rate) * stickiness_state +
          stickiness * (1 - reward_noisy) * jax.nn.one_hot(choice, 2))

    choice_logits = inv_temp * (
          q_values_updated + stickiness_state +
          bias * jnp.array([1.0, -1.0]))
    exploration_term = (
          exploration_rate * jnp.exp(-trial_count / 20) * jnp.array([q_values_updated[1],
          q_values_updated[0]]))
    choice_logits = choice_logits + exploration_term
    updated_lapse_rate = (
          (1 - lapse_alpha) * lapse_rate_state +
          lapse_alpha * jnp.clip(lapse_rate, 0.001, 0.999))
    p = (
          (1 - updated_lapse_rate) * jax.nn.softmax(choice_logits) +
          updated_lapse_rate * jnp.array([0.5, 0.5]))
```

```python
  p = jnp.clip(p, 1e-8, 1 - 1e-8)
  choice_logits = jnp.log(p / (1 - p))

  q_values = momentum * q_values_updated + (1 - momentum) * q_values
  q_values = (1 - decay_rate) * q_values + decay_rate * init_q

  trial_count += 1

  agent_state = jnp.concatenate(
      [q_values, stickiness_state,
       lapse_rate_state[None],
       trial_count[None]])

  return choice_logits, agent_state
```

## H.4. Rat FullModel

Best FullModel program given a cutoff of 165,000 samples.

```python
def agent(params, choice, reward, agent_state = None):
  alphas = jax.nn.sigmoid(params[:5]) + 0.05
  betas = params[5:10]
  bias_term = params[10]
  tau = jax.nn.softplus(params[11]) + 1.
  lapse_rate = jax.nn.sigmoid(params[12]) * 0.25 + 0.0

  if agent_state is None:
    agent_state = jnp.array([0., 0., 0., 0., 0.5, 0.5])

  choice_for_update = 2 * choice - 1
  reward_for_update = 2 * reward - 1

  reward_prediction_error = reward_for_update - agent_state[4]

  agent_state = agent_state.at[0].set(
      alphas[0] * agent_state[0] +
      (1 - alphas[0]) * reward_prediction_error *
      choice_for_update * (1 - agent_state[5])
  )
  agent_state = agent_state.at[1].set(
      alphas[1] * agent_state[1] +
      (1 - alphas[1]) * choice_for_update
  )
  agent_state = agent_state.at[2].set(
      alphas[2] * agent_state[2] + (1 - alphas[2]) * (
          reward_for_update * choice_for_update -
          choice_for_update)
  )
  agent_state = agent_state.at[3].set(
      alphas[3] * agent_state[3] +
      (1 - alphas[3]) * reward_prediction_error * (1 - agent_state[5])
  )
  agent_state = agent_state.at[4].set(
      alphas[4] * agent_state[4] +
      (1 - alphas[4]) * reward_prediction_error * (1 - agent_state[5])
  )
  agent_state = agent_state.at[5].set(
      alphas[4] * agent_state[5] +
      (1 - alphas[4]) * reward_prediction_error * (1-agent_state[5])
  )

  choice_term = (
      betas[0] * agent_state[0] +
      betas[1] * agent_state[1] +
      betas[2] * agent_state[2] +
      betas[3] * agent_state[3] +
      betas[4] * agent_state[4] +
      bias_term) / tau

  #choice_term = (betas[0] * agent_state[0] + betas[1] * agent_state[1] +
  #betas[2] * agent_state[2] + betas[3] * agent_state[3] +
  #betas[4] * jnp.maximum(agent_state[4], 0) + bias_term) / tau

  choice_logits = (1 - lapse_rate) * jnp.array(
      [choice_term, -choice_term]) + lapse_rate * jnp.log(jnp.array([0.5, 0.5]))
  return choice_logits, agent_state
```

# I. Example Programs that vary in Complexity

These are all programs sampled from the complexity/accuracy frontier, as discussed in Section 4.2 and illustrated in Figures 7, 13.

## I.1. Human Example Programs

Here we share the Example Programs for the Human dataset, indicated in Fig. 13. These programs are ordered from least complex / lowest scoring to most complex / highest scoring, along the frontier. We provide a brief description of the code's function, for convenience. These programs were all generated with the Structured2 seed program, which also generated the best Human CogFunSearch programs.

**Example Program 1.** This program substantially overlaps with the PFQ baseline, exhibiting Q-learning and forgetting. Interestingly, it does not exhibit any perseveration (it records previous choice in its state, but never uses it). The key addition in this program is optimistic initialization in which all Q-values start at 1.

```python
def agent(params, choice, reward, agent_state):

  if agent_state is None:
    agent_state = jnp.ones((5,)) # start all Q-values at 1

  beta_r = jnp.exp(params[0])
  alpha_learn = jax.nn.softplus(params[1])
  alpha_forget = jax.nn.softplus(params[2])

  q_values = agent_state[:4]

  # Update Q values
  updated_qvalues = q_values * (1 - alpha_learn) + alpha_learn * reward
  updated_qvalues = updated_qvalues.at[choice].set(
      updated_qvalues[choice] +
      alpha_learn * (reward - q_values[choice]))

  # Decay Q values
  decaying_qvalues = updated_qvalues * jnp.exp(-alpha_forget)

  # Compute choice probabilities using softmax
  choice_logits = beta_r * decaying_qvalues

  # Update agent state.
  new_agent_state = jnp.concatenate([decaying_qvalues, jnp.array([choice])])

  return choice_logits, new_agent_state
```

**Example Program 2.** Example Program 2 is similar to 1, except that initial Q-values are now a fittable parameter instead of optimistically initialized to 1, and forgetting decays toward 0.5 rather than 0 (as in Example Program 1) or the mean Q-value (as in the best CogFunSearch program). This program introduces an additional parameter compared to Example Program 1, **lapse_rate**, which is also found in the highest scoring CogFunSearch runs and evokes the classic psychology literature.

```python
def agent(params, choice, reward, agent_state):

  # Define parameters
  beta_r = jax.nn.softplus(params[0])
  alpha_learn = jax.nn.sigmoid(params[1])
  alpha_forget = jax.nn.sigmoid(params[2])
  lapse_rate = jax.nn.sigmoid(params[3])

  initial_q = jax.nn.sigmoid(params[4])  # Initialize Q values close to 0.5

  #Initialize state
  if agent_state is None:
    agent_state = jnp.repeat(initial_q, 4)

  q_vals = agent_state
```

```
# Update Q-values
prediction_error = reward - q_vals[choice]
q_vals = q_vals.at[choice].set(
    (1 - alpha_learn) * q_vals[choice] + alpha_learn * reward
)
# Forgetting process
q_vals = q_vals * (1 - alpha_forget) + 0.5 * alpha_forget # Adds in forgetting that favors 0.5

# Incorporate lapse rate
choice_probs = (
    (1 - lapse_rate) * jax.nn.softmax(beta_r * q_vals) +
    lapse_rate * (1.0 / 4.0) * jnp.ones(4))
choice_logits = jnp.log(choice_probs)

return choice_logits, q_vals
```

**Example Program 3.** is considerably higher scoring and more complex Figure 13.

Like Example Program 2, this program also includes fittable initial Q-values. Both programs also track previous state, but don't actually use it to inform any computation. The practice of decaying Q-values of chosen actions toward the mean Q-value, which is present in all of the top CogFunSearch programs, is also introduced in this program. This program uses recent reward to modulate exploration, with more exploitation if the immediately preceding trial was rewarded.

```
def agent(params, choice, reward, agent_state):

    param_names = ['beta_r', 'alpha_learn', 'alpha_forget', 'omega', 'inverse_temperature', 'init_q']
    params_dict = dict(zip(param_names, params))

    if agent_state is None:
        # Default values
        agent_state = jnp.ones(5) * 0 # Initialize
        agent_state = agent_state.at[:4].set(jax.nn.softplus(params_dict['init_q'])) # Set the Q values
        agent_state = agent_state.at[4].set(-1)

    #Define parameters
    beta_r = jax.nn.softplus(params_dict['beta_r']) + 0.1
    alpha_learn = jax.nn.sigmoid(params_dict['alpha_learn'])
    alpha_forget = jax.nn.sigmoid(params_dict['alpha_forget'])
    omega = jax.nn.softplus(params_dict['omega'])
    inverse_temperature = jax.nn.softplus(params_dict['inverse_temperature']) + 0.01
    init_q = jax.nn.softplus(params_dict['init_q'])

    # Update agent state
    q_values = agent_state[:4]
    previous_choice = agent_state[4]

    q_values = q_values.at[choice].set(
        q_values[choice] + alpha_learn * (reward - q_values[choice])) # Update Q value for chosen arm

    alpha_forget_mult = 1.0 - alpha_forget
    q_values = (
        q_values * alpha_forget_mult +
        alpha_forget * jnp.mean(q_values)) # Apply forgetting weighted by overall mean Q value

    #Compute choice logits
    # Apply inverse temperature and reward modulation
    choice_logits = beta_r * q_values / inverse_temperature * (1 + omega * reward**2)
    choice_probabilities = jax.nn.softmax(
        choice_logits,axis = 0)   #Apply softmax for probabilities

    choice_logits = jnp.log(choice_probabilities + 1e-8)

    final_state = jnp.concatenate(
        (q_values, jnp.array([choice]))) # Update the final state

    return choice_logits, final_state
```

## I.2. Rat Example Programs

Example Programs for the Rat dataset, indicated in Figs. 7, 13. These programs are ordered from least complex / lowest scoring to most complex / highest scoring, along the frontier. We provide a brief description of the code's function, for

convenience. These programs were all generated with the FullModel seed program, which also generated the best Rat CogFunSearch programs.

**Example Program 1.** This program preserves the terms R, H, and G, expanding them to `reward`, `habit`, and `guess`. It is notable that the LLM maps `H` in the seed program to `habit`, as the word habit is not included anywhere in the seed program or initial prompt documentation. It maps `G` to `guess`, which is different from the RHG baseline where it refers to gambler's fallacy. This program has only 5 parameters, compared to the RHG's model's seven, and the same learning rate is used across all state updates. The code shows a similar update to RHG, except instead of tracking the recently rewarded choice, it simply tracks recent reward. Then, all state variables are summed.

```python
def agent(params, choice, reward, agent_state):

  if agent_state is None:
    agent_state = jnp.zeros((3,))

  v_reward = params[0]
  v_habit = params[1]
  v_guess = params[2]
  bias = params[3]
  learning_rate = jax.nn.sigmoid(params[4])

  prev_choice = 2 * choice - 1
  prev_reward = 2 * reward - 1

  # Update reward prediction.
  agent_state = agent_state.at[0].set(
      (1 - learning_rate) * agent_state[0] + learning_rate * prev_reward
  )

  # Update choice value.
  agent_state = agent_state.at[1].set(
      (1 - learning_rate) * agent_state[1] + learning_rate * prev_choice
  )

  # Update guessing value.
  agent_state = agent_state.at[2].set(
      (1 - learning_rate) * agent_state[2] +
      learning_rate * (prev_choice - prev_reward * prev_choice)
  )

  # Compute choice logits.
  choice_term = (
  v_reward * agent_state[0] + v_habit * agent_state[1] + v_guess * agent_state[2] +
  bias)
  choice_logits = jnp.array([-1, 1]) * choice_term

  return choice_logits, agent_state
```

**Example Program 2.** uses the same parameters as the RHG baseline, plus an additional temperature parameter controlling exploration. It also defines 7 additional state variables that separately track reward and omission rates, as well as a complex interaction of reward and choice.

```python
def agent(params, choice, reward, agent_state):

  n_params = params.shape[0]
  if agent_state is None:
    agent_state = jnp.zeros((10,))

  alpha_r = jax.lax.clamp(0.001, 0.999, jax.nn.sigmoid(params[0]))
  alpha_h = jax.lax.clamp(0.001, 0.999, jax.nn.sigmoid(params[1]))
  alpha_g = jax.lax.clamp(0.001, 0.999, jax.nn.sigmoid(params[2]))
  beta_r = jax.nn.softplus(params[3])
  beta_h = jax.nn.softplus(params[4])
  beta_g = jax.nn.softplus(params[5])
  bias_params = params[6:8]  # bias is now a vector
  inverse_temperature = jax.nn.softplus(params[8])

  choice_for_update = 2 * choice - 1
  reward_for_update = 2 * reward - 1

  delta_r = reward_for_update * choice_for_update
  agent_state = agent_state.at[0].set(alpha_r * agent_state[0] + (1 - alpha_r) * delta_r)
```

```
    delta_h = choice_for_update
    agent_state = agent_state.at[1].set(alpha_h * agent_state[1] + (1 - alpha_h) * delta_h)
    delta_g = choice_for_update - reward_for_update * choice_for_update
    agent_state = agent_state.at[2].set(alpha_g * agent_state[2] + (1 - alpha_g) * delta_g)

    # Separate parameters for biases.
    # Exponential decay, with a dynamic decay rate depending on the reward expectation.
    decay_rate_for_reward = jnp.clip(0.2 + 0.5*jnp.tanh(agent_state[6]), 0.1, 0.9)
    decay_rate_for_punishment = jnp.clip(0.5 + 0.5*jnp.tanh(agent_state[7]), 0.1, 0.9)

    agent_state = agent_state.at[3].set(
        decay_rate_for_reward * (
            agent_state[3] + reward_for_update * choice_for_update * bias_params[0])
    ) # Bias for choice 1
    agent_state = agent_state.at[4].set(
        decay_rate_for_punishment * (
            agent_state[4] + reward_for_update * choice_for_update * bias_params[1])
    )  # Bias for choice -1
    agent_state = agent_state.at[6].set(
        0.99 * agent_state[6] + 0.01 * reward_for_update)
    agent_state = agent_state.at[7].set(
        0.99 * agent_state[7] + 0.01 * (1-reward_for_update))

    state_1 = agent_state[3] + agent_state[4]   #changed line

    choice_term = (beta_r * agent_state[0] + beta_h * agent_state[1] +
                   beta_g * agent_state[2] + state_1 + params[-1])
    choice_logits = jnp.array([-1, 1]) * choice_term * inverse_temperature

    return choice_logits, agent_state
```

**Example Program 3.** This program has very similar code to the RHG baseline. It includes a different parameterization of the parameters, which constrains all learning rates to be clamped at 0.01 and 0.99, lacks a bias term, and includes a step where tanh applied to the choice term. Charmingly, this tanh step is accompanied by a comment claiming: "This is different from the other version, potentially an important improvement". Verifying this claim would require future investigation.

```
def agent(params, choice, reward, agent_state):

    if agent_state is None:
        agent_state = jnp.zeros((3,))

    alpha_r = jax.lax.clamp(0.01, 0.99, jax.nn.softplus(params[0]))
    alpha_h = jax.lax.clamp(0.01, 0.99, jax.nn.softplus(params[1]))
    alpha_g = jax.lax.clamp(0.01, 0.99, jax.nn.softplus(params[2]))
    beta_r = jax.nn.softplus(params[3])
    beta_h = jax.nn.softplus(params[4])
    beta_g = jax.nn.softplus(params[5])

    # Convert choice and reward 0/1 to +-1
    choice_for_update = 2 * choice - 1
    reward_for_update = 2 * reward - 1

    # Update of values R, H, G.
    delta_r = reward_for_update * choice_for_update
    agent_state = agent_state.at[0].set(alpha_r * agent_state[0] + (1 - alpha_r) * delta_r)
    delta_h = choice_for_update
    agent_state = agent_state.at[1].set(alpha_h * agent_state[1] + (1 - alpha_h) * delta_h)
    delta_g = choice_for_update - reward_for_update * choice_for_update
    agent_state = agent_state.at[2].set(alpha_g * agent_state[2] + (1 - alpha_g) * delta_g)

    # This is different from the other versions, potentially an important improvement.
    choice_term = jnp.tanh(
        beta_r * agent_state[0] +
        beta_h * agent_state[1] +
        beta_g * agent_state[2]) +
        params[6] +
        params[7] * choice_for_update +
        params[8] * reward_for_update
    )

    choice_logits = jnp.array([-1, 1]) * choice_term * 2 # this is a better softmax
    return choice_logits, agent_state
```

## I.3. Fruit Fly Example Programs

Example Programs for the Fruit Fly dataset, indicated in Figure 13. These programs are ordered from least complex / lowest scoring to most complex / highest scoring, along the frontier. We provide a brief description of the code's function, for convenience. These programs were all generated with the FullModel seed program, which also generated the best Rat CogFunSearch programs.

**Example Program 1.** This program closely imitates the DFQ baseline, adding a bias toward which the Q-value of the unchosen action decays.

```python
def agent(params, choice, reward, agent_state):

  if agent_state is None:
    agent_state = jnp.array([0.0, 0.0])

  # Unpack and rename parameters for readability.
  alpha_learn = jax.nn.softplus(params[0])   # Ensures alpha_learn is positive
  alpha_forget = jax.nn.softplus(params[1])   # Ensures alpha_forget is positive
  kappa_reward = jax.nn.softplus(params[2])   # Ensures kappa_reward is positive
  kappa_omission = jax.nn.softplus(params[3])   # Ensures kappa_omission is positive
  bias = params[4]
  qs = agent_state

  # Chosen action's Q-value will be updated towards this target value.
  learn_target = kappa_reward * reward + kappa_omission * (1 - reward)

  # Update Q-value for chosen action.
  new_q = qs[choice] + alpha_learn * (learn_target - qs[choice])
  qs = qs.at[choice].set(new_q)
  # Update Q-value for unchosen action: decay towards bias.
  qs = qs.at[1 - choice].set(qs[1 - choice] - alpha_forget * (qs[1 - choice] - bias))

  # Add a bias to account for initial preference.
  logits = jax.nn.softplus(qs + bias)
  new_agent_state = qs
  return logits, new_agent_state
```

**Example Program 2.** This program closely imitates the DFQ baseline, adding a bias toward which the Q-value of the unchosen action decays. Like other discovered programs, including the best CogFunSearch Fruit Fly program, it includes a **lapse_rate** term. Like the best CogFunSearch program, this program includes a somewhat unusually complex exploration policy.

```python
def agent(params, choice, reward, agent_state):

  if agent_state is None:
    agent_state = jnp.array((0.5, 0.5))

  # Unpack parameters
  alpha_learn_logit = params[0]
  beta_logit = params[1]
  kappa_reward_logit = params[2]
  kappa_omission_logit = params[3]
  tau_logit = params[4]
  lapse_rate_logit = params[5]
  alpha_forget_logit = params[6]

  alpha_learn = jax.nn.softplus(alpha_learn_logit)
  beta = jnp.exp(beta_logit)   # enforce beta > 0
  kappa_reward = jax.nn.softplus(kappa_reward_logit)
  kappa_omission = jax.nn.softplus(kappa_omission_logit)
  tau = jax.nn.softplus(tau_logit)
  lapse_rate = jax.nn.sigmoid(lapse_rate_logit)
  alpha_forget = jax.nn.softplus(alpha_forget_logit)

  qs = agent_state

  # update q value.
  learn_target = kappa_reward * reward + kappa_omission * (1 - reward)
  error = (learn_target - qs[choice])

  qs = qs.at[choice].set(qs[choice] + alpha_learn * error)
  qs = qs.at[1 - choice].set(qs[1 - choice] - alpha_forget * error)
```

```
qs = jnp.clip(qs, 0., 1.)
qs = qs + 0.001

# add lapse rate, ensure that lapse_rate is not 1.0
logit = (1. - lapse_rate) * tau * (qs[0] ** beta - qs[1] ** beta) + lapse_rate * jnp.log(0.5)

new_agent_state = qs
choice_logits = jnp.array([logit, -logit])
return choice_logits, new_agent_state
```

**Example Program 3.** This program again includes the functionality from the DFQ baseline, and in fact closely resembles Example Program 1. It also includes an unusual, dynamically updated exploration term which aggregates the difference between the reward and the lapse rate, and uses this to scale eloration.

```
def agent(params, choice, reward, agent_state):

  if agent_state is None:
    agent_state = jnp.array((0.5, 0.5, 0., 0.))

  # Unpack parameters
  alpha_learn_logit = params[0]
  beta_logit = params[1]
  kappa_reward_logit = params[2]
  kappa_omission_logit = params[3]
  tau_logit = params[4]
  lapse_rate_logit = params[5]
  alpha_forget_logit = params[6]
  alpha_lapse_logit = params[7]

  alpha_learn = jax.nn.softplus(alpha_learn_logit)
  beta = jnp.exp(beta_logit)  # enforce beta > 0
  kappa_reward = jax.nn.softplus(kappa_reward_logit)
  kappa_omission = jax.nn.softplus(kappa_omission_logit)
  tau = jax.nn.softplus(tau_logit)
  lapse_rate = jax.nn.sigmoid(lapse_rate_logit)
  alpha_forget = jax.nn.softplus(alpha_forget_logit)
  alpha_lapse = jax.nn.softplus(alpha_lapse_logit)

  qs = agent_state[:2]
  lapse = agent_state[2]
  lapse_rate_ = agent_state[3]

  # update q value.
  learn_target = kappa_reward * reward + kappa_omission * (1 - reward)
  error = (learn_target - qs[choice])

  qs = qs.at[choice].set(qs[choice] + alpha_learn * error)
  qs = qs.at[1 - choice].set(qs[1 - choice] - alpha_forget * error)

  qs = jnp.clip(qs, 0., 1.)
  qs = qs + 0.001

  # update lapse rate.
  lapse = lapse + alpha_lapse * (reward - lapse)

  # add lapse rate, ensure that lapse_rate is not 1.0
  logit = (1. - lapse) * tau * (qs[0] ** beta - qs[1] ** beta) + lapse_rate_ * jnp.log(0.5)

  new_agent_state = jnp.array([qs[0], qs[1], lapse, lapse_rate])
  choice_logits = jnp.array([logit, -logit])
  return choice_logits, new_agent_state
```

