# OpenReview forum: "Discovering Symbolic Cognitive Models from Human and Animal Behavior"
_ICML.cc/2025/Conference — ICML 2025 spotlightposter_

### Official Review · Reviewer_wzmy · 2025-03-09

**Overall Recommendation:** 4

**Summary:**

This paper presents a new method, CogFunSearch, that automatically discovers symbolic models for a given dataset. Their approach builds on FunSearch [1], an LLM-driven evolutionary algorithm that searches over the program’s structure, by adding an inner level of optimization that fits model parameters to the data. Experiments on behavioral datasets from humans, rats, and fruit flies demonstrates that CogFunSearch discovers novel symbolic models that outperform hand-engineered programs and are human-interpretable.

[1] Romera-Paredes et al. “Mathematical discoveries from program search with large language models” (2023).

**Claims And Evidence:**

Yes.

**Essential References Not Discussed:**

The key contribution in terms of methods is augmenting FunSearch with bilevel optimization over program structure and parameters. I believe that the paper is missing a reference to [1], which proposes a differentiable inner-level optimization with a LLM-driven evolutionary search.

[1] Ma et al. “LLM and Simulation as Bilevel Optimizers: A New Paradigm to Advance Physical Scientific Discovery” (2024).

**Experimental Designs Or Analyses:**

Yes.

**Methods And Evaluation Criteria:**

Yes, I checked the experiments in the main paper and the evaluation details in Appendix B.

**Other Comments Or Suggestions:**

1. Missing period at the end of Figure 6.
2. I would move Figure 1 closer to the results.

**Other Strengths And Weaknesses:**

Strengths
1. The application of LLM-driven program synthesis to cognitive modeling is novel.
2. Experiments demonstrate that the best discovered programs achieve better performance than a hand-engineered baseline.
3. Analysis of the complexity and readability of the outputted programs demonstrates that the discovered programs yield new behavioral insights.

Weaknesses
1. As the inner optimization loop fits parameters using gradient descent, it requires the programs to be differentiable, which may hinder the application of the approach to other settings.
2. I’m a bit unsure how to interpret the complexity scores, as in how do the relative complexity scores compare to a human expert judgement.
3. CogFunSearch requires a significant amount of compute, although the paper explores rejection sampling as a way to improve the efficiency.
4. While more interpretable than a black-box network, the symbolic models may contain errors or be overly complex, hindering readability.

**Questions For Authors:**

1. Are the set of parameters the same for all the programs? If so, how does one decide which parameters to use?
2. How do the complexity scores align with a human expert’s perception of the complexity? What is the optimal tradeoff between complexity and performance?
3. How data-efficient is CogFunSearch compared to the RNN? For example, would CogFunSearch outperform the RNN if trained on only 10% of the subjects?

**Relation To Broader Scientific Literature:**

There exists some prior work that combine LLM-drive search over program structure with an inner parameter fitting step [1, 2]. As such, the main novelty of the paper is not in the method but instead the application to automating the discovery of symbolic cognitive models, which are typically hand-engineered by domain experts.

[1] Ma et al. “LLM and Simulation as Bilevel Optimizers: A New Paradigm to Advance Physical Scientific Discovery” (2024).

[2] Li et al. “Automated Statistical Model Discovery with Language Models” (2024).

**Theoretical Claims:**

N/A.

---

> ### Author Rebuttal · Authors · 2025-03-31
>
> We thank the reviewer for their careful review of our submission, their useful comments, and are glad they found our approach “novel” and yielding “new behavioral insights”.
>
> ## “missing reference to [1].”
>
> Thank you for pointing out this relevant work! We will cite it and include it in our discussion.
>
> ## W1. “requiring programs to be differentiable may hinder the application to other settings.”
>
> This is true. We considered it a reasonable limitation, given that the most popular models for these datasets are all differentiable. One could in principle use a gradient-free or hybrid  optimizer for the inner loop (e.g. Acerbi & Ma, 2017), and this would expand the space of models that could be considered. However, this would affect the computational cost required for scoring the models.  We will revise the manuscript to be clear about this limitation.
>
> ## W2 & Q2a. “how do the relative complexity scores compare to a human expert judgement.”
>
> Automatic complexity of programs is an open problem with multiple proposed solutions in the literature, which is why we use multiple methods. Halstead Difficulty is a classic and widely-used measure designed to capture the difficulty of constructing or parsing the program. Several studies do exist linking it to behavioral measures and subjective complexity (e.g. Curtis et al., 2006; De Silva et al., 2012; Gao et al., 2025), but to our knowledge it has not been thoroughly evaluated for Python.  LLMs, having been trained on human data and optimized for human tasks, can also be considered a proxy for human judgments, though this approach also lacks thorough validation. For our programs, we find that these scores largely align (as shown in Figs 7, 12, 13).
>
> Thoroughly validating these measures ourselves was beyond the scope of this work, but we do include example programs in the supplement so that readers can assess how they compare to their own intuitive notions of complexity.
>
> ## W3. “CogFunSearch requires a significant amount of compute”
>
> While this is true, it is not clear how to compare compute cost appropriately against alternative approaches. The baselines models considered represent the results of years of human effort, and were therefore clearly vastly more expensive to produce than the programs we generate automatically. We expect that advances in hardware, model efficiency, and alternative evolution techniques (our rejection sampling being a simple example of one) will result in reduced computational costs for applying approaches such as CogFunSearch.
>
> ## W4. “symbolic models may contain errors or be overly complex, hindering readability.”
>
> While discovered programs showed sufficient interpretability/readability to afford insights about learning strategies, it is true that there is room for improvement. Exploring tools for minimizing extraneous complexity or improving the readability of the discovered programs is a compelling direction for future work.
>
> It is also worth mentioning that while the best-fitting models were more complex than human-discovered baselines, there were many discovered programs that were less complex but still outperformed the baselines.
>
> ## Q1. “Are the set of parameters the same for all the programs? If so, how does one decide which parameters to use?”
>
> All programs are provided with the same number of free parameters (10), but each program will use these differently (and some may even use fewer than 10).
>
> As a concrete example: param[0] may be `learning_rate` for one program, and `lapse_rate` in another. Furthermore, each subject in a dataset can have its own value for each parameter, so `learning_rate` for rat 1 may be different than learning rate for rat 13, but the parameter will be used in the same way.
>
> ## Q2b. “ optimal tradeoff between complexity and performance?”
>
> This would depend on what the model will be used for. In some use cases, greater interpretability (possibly at the expense of accuracy) may be more desirable, and in others, the opposite may hold. One of the advantages of our approach is that it produces multiple programs along this efficiency frontier, enabling researchers to identify the program(s) that are useful to them. We expect a major use case for the tool will be to surface useful ideas about how to model the dataset, & that the best approach will be to examine multiple programs at different points along the complexity/performance tradeoff.
>
> ## “How data-efficient is CogFunSearch compared to the RNN? For example, would CogFunSearch outperform the RNN if trained on only 10% of the subjects?”
>
> While we have not run this experiment directly, our comparison on the human dataset, where RNN struggles with per-subject parameters, is an indication that CogFunSearch is able to find more generalizable solutions that are still accurate.

---

> > ### Comment · Reviewer_wzmy · 2025-04-02
> >
> > I thank the authors for the clarification to my questions. I maintain my current score of recommending the paper for acceptance.

---

### Official Review · Reviewer_fWgP · 2025-03-09

**Overall Recommendation:** 4

**Summary:**

This paper proposes to extend FunSearch (Romera-Paredes et al., 2024) to symbolic cognitive modeling, namely CogFunSearch, an LLM-based evolutionary program synthesis framework.
Experimental results have strongly supported the value of CogFunSearch in discovering high-quality symbolic programs on human, rat, and fruitfly bandit datasets, outperforming human-designed symbolic models proposed in very recent literature.

**Claims And Evidence:**

In what follows, I will use S, W, C, and Q to denote Strengths, Weaknesses, Comments, and Questions, respectively, for ease of reference in the discussion.

The authors propose CogFunSearch, an LLM-based symbolic cognitive modeling framework, and demonstrate the effectiveness of the proposed framework on three bandit datasets of human, rat, and fruitfly behaviors. Results have clearly supported that the proposed framework can discover high-quality symbolic cognitive models that outperform human-designed symbolic models.

S1. The paper is exceptionally well-written and (surprisingly) easy to follow. Although I am not an expert in the specific cognitive modeling task addressed in this paper, the authors have done an excellent job in describing the data and formulating the problem, so I can understand without difficulty.

S2. The authors have made a proper yet exciting claim---the authors did not claim "LLMs can substitute cognitive scientists" but rather focused on the specific problem of symbolic cognitive modeling, and the experiments have shown convincing results that the discovered models are of high quality.

**Essential References Not Discussed:**

C2. This is not a mandatory request, but connecting to the work that uses LLMs to propose hypotheses in other CS-related areas would be helpful to provide a comprehensive background to the general ICML audience. For example, you may wish to check out https://arxiv.org/abs/2408.05086 and https://aclanthology.org/2024.nlp4science-1.10.pdf.

**Experimental Designs Or Analyses:**

As mentioned in the Methods and Evaluation Criteria section, the main experiments are around comparing model likelihood on held-out data, a common metric in similar machine-learning tasks. The authors have also compared RNNs and LLM-based program synthesis, in terms of the gap between each of them and the human-designed baseline models.

W1. The authors only evaluates Gemini 1.5 Flash---which is a considerably strong model but not among the best ones according to my machine-learning knowledge (in my area(s), Gemini 1.5 Pro, OpenAI o1, Anthropic Claude models, and DeepSeek models are usually considered the best ones at the current stage)---as the backend LLM for program synthesis. From a machine-learning perspective, the paper could be made stronger if more LLMs are tested.

W2. For RNN results (Figure 5), it's unclear (1) whether the authors trained the RNNs themselves or they used others' trained RNN models, (2) what the specific RNN architecture they used, (3) what hyperparameters they tuned to optimize the RNNs. (3) is particularly important as given the data size, both overfitting and underfitting are possible, and the results could be very sensitive to hyperparameters. I would recommend the authors pay special attention to this part.

C1. Related to above, it would also be good to cite the original work of RNN (Elman, 1990), and, if you used a specific RNN architecture (such as LSTM), please also cite the corresponding paper(s).

S4. This paper has a strong discussion on the qualitative results of the discovered symbolic models (Section 4), which highlight the importance of the proposed framework in scientific discovery.

**Methods And Evaluation Criteria:**

The method used in this paper is basically an extension of FunSearch (Romera-Paredes et al., 2024) to symbolic cognitive modeling.
- The main evaluation criterion to compare models is the likelihood on the held-out data, which is a common metric in machine learning tasks with similar settings and purposes.
- Additionally, the authors consider the quality of synthesized data, in terms of reproducing scientifically important features, as a secondary metric to illustrate model quality---qualitatively, the discovered programs have shown a sufficiently similar pattern to collected animal data.
- Finally, the authors have examined the strong robustness of their results with respect to random seeds.

S3. I endorse that the above evaluation criteria form a comprehensive and convincing evaluation protocol for the proposed framework.

**Other Comments Or Suggestions:**

C3. The bibliography needs some cleaning.
- There is an incomplete "Palminteri. 2015." entry in the reference.

C4. Some presentation suggestions and typos:
- You may wish to consider making the average lines in Figures 1 and 5 thinner, or consider using a box plot for it. I can only realize what the "blue rectangles" in Figure 1 left & right are by generalizing from the middle.

- L649: "Fig B.1." has pointed to Section B.1. You may wish to fix this.

**Other Strengths And Weaknesses:**

N/A. I've discussed all strengths and weaknesses I found in the above sections.

**Questions For Authors:**

Q1. Is Figure 2B necessarily a Bayesian program synthesis pipeline? It seems that the "prior" $\theta$ is not necessarily a distribution, but a point estimate. Is there any obstacle to generalize it to a distribution?

Q2. L155 (right): Why is $c\in \{0, \ldots, n\}$ instead of $c\in \{1, \ldots, n\}$? Is it allowed to have a "null" choice? If the latter is the case, you may wish to clarify, as it's not a common bandit problem formulation.

Q3. If I understood correctly, CogFunSearch is a general framework and should work for general data science tasks that involve symbolic modeling. Are there anything specific things about cognitive modeling that make CogFunSearch particularly suitable for it?

C5. Finally, I would like to thank the authors for the strong submission. I enjoyed reviewing the paper and learnt a lot.

**Relation To Broader Scientific Literature:**

W3. It is notable that the authors included the related work discussion in supplementary material. I would strongly recommend moving it (or at least the most important ones) to the main text, as it is crucial for grounding the proposed method to existing literature.

**Theoretical Claims:**

This paper does not contain much theoretical claim, but rather proposes a practical framework for symbolic cognitive modeling using LLMs. However, there could be some interesting theoretical implications of the proposed framework around the interpretability of LLMs and the benefits of continuous vs. discrete representations in cognitive modeling.

---

> ### Author Rebuttal · Authors · 2025-03-31
>
> We thank the reviewer for their careful review of our submission, their useful comments, and are glad they found it “exceptionally well-written and (surprisingly) easy to follow”, and are thrilled (and grateful) they “enjoyed reviewing the paper and learnt a lot”.
> Below, we address the main issues raised.
>
> ## W1. “The authors only evaluate Gemini 1.5 Flash… the paper could be made stronger if more LLMs are tested.”
>
> Our main intent with this work was to demonstrate that the use of LLMs with CogFunSearch was capable of producing state of the art results. An important consideration in methods that use LLMs for the mutation function in an evolutionary algorithm is the tradeoff between model quality (relating to expected performance improvement per sample) and model throughput (relating to the number of samples that can practically be drawn). We chose Gemini 1.5 Flash specifically because it was designed to provide a reasonable tradeoff between these concerns, rather than providing the absolute best quality samples. Although we did not explore the use of other LLMs systematically, we agree this would be a natural idea for pushing the performance of this approach even further.
>
> ## W2. “For RNN results it's unclear (1) whether the authors trained the RNNs themselves, (2) what architecture they used, (3) what hyperparameters they tuned.”
>
> We trained our own RNNs, using  a GRU model (Chung et al., 2014) with a single hidden layer, and trained with early stopping. We performed a sweep over a set of possible hidden units (`[1, 2, 4, 8, 16, 32, 64, 128]`) and picked the value that gave the best performance. All the variants were trained with the Adam optimizer [(Kingma & Ba, 2015)](https://arxiv.org/abs/1412.6980) with a learning rate of `1e-3`.
>
> Thank you for raising this point, as we inadvertently missed including these details, and will be including them in the revised version of the paper.
>
> ## W3. “recommend moving related work discussion to the main text.”
>
> We agree, and will be following your suggestion to move the related work (or the most important parts of it) to the main paper.
>
> ## C2. “connecting to the work that uses LLMs to propose hypotheses in other CS-related areas would be helpful to provide a comprehensive background to the general ICML audience”
>
> Thank you for this suggestion and for the provided references. We agree it would be valuable to connect to this body of related work, and will be including it in our revised related work section.
>
> ## C3 and C4
>
> We will address these issues, thank you for pointing them out.
>
> ## Q1. “Is Figure 2B necessarily a Bayesian program synthesis pipeline? It seems that the "prior" $\theta$ is not necessarily a distribution, but a point estimate. Is there any obstacle to generalize it to a distribution?”
>
> We would like to clarify that $\theta$ is not a prior, but rather the parameters that are fit in the internal optimization process (Fig. 2). However, we do essentially maintain a "distribution" over programs, via our program database (Fig 2A). The prior distribution over these programs is governed by the score $\Omega$ of each program, which is proportional to the likelihood of it being sampled for evolution. When a new program is generated, this "prior" distribution is updated as a new program (with its score $\Omega$) is added to the database.
>
> ## Q2. “L155 (right): Why is $c \in 0, \ldots ,n$ instead of $c \in 1, \ldots ,n$? Is it allowed to have a "null" choice?”
>
> Thank you for pointing this out, it should be $c \in 1, \ldots ,n$, as the reviewer suspected. We will correct this in the revised version.
>
> ## Q3. “If I understood correctly, CogFunSearch is a general framework and should work for general data science tasks that involve symbolic modeling. Are there anything specific things about cognitive modeling that make CogFunSearch particularly suitable for it?”
>
> Our bilevel optimization arose from a common modeling choice in cognitive science, which is that programs capture across-subject patterns and parameters capture individual variations. For example, a cognitive modeling setup might assume or hypothesize that all subjects are executing the same RL algorithms, but each subject has a different learning rate and exploration parameter. In our case, CogFunSearch is set up to fit unique per-subject parameters in the inner loop, and programs are evolved based on the average across-subject score in the outer loop.
>
> However, as the reviewer suggests, this can be applicable to other situations with similar setups. The core of our approach is using FunSearch to discover parameterized programs that can be fit to a dataset, which is applicable to other areas of data-driven scientific discovery.

---

> > ### Comment · Reviewer_fWgP · 2025-04-01
> >
> > I thank the authors for the careful response and am now at the highest possible level of confidence to recommend it for a clear acceptance.
> >
> > One minor thing from the rebuttal though: GRU is from Cho et al. (2014) and not Chung et al.

---

> > > ### Author Response · Authors · 2025-04-02
> > >
> > > We thank the reviewer for pointing this out, and we will correct the citation to the one below.
> > >
> > > _Kyunghyun Cho, Bart van Merriënboer, Dzmitry Bahdanau, and Yoshua Bengio_. 2014. **On the Properties of Neural Machine Translation: Encoder–Decoder Approaches**. In Proceedings of SSST-8, Eighth Workshop on Syntax, Semantics and Structure in Statistical Translation, pages 103–111, Doha, Qatar. Association for Computational Linguistics.

---

### Official Review · Reviewer_nJ74 · 2025-03-12

**Overall Recommendation:** 3

**Summary:**

This paper extends the FunSearch evolutionary algorithm to autonomously uncover symbolic cognitive models that effectively represent human and animal behavior. The authors compare the top-discovered program with an RNN trained on data from all subjects collectively, showcasing the efficacy of CogFunSearch. Additionally, they explore the trade-off between quality of fit and program complexity.

**Claims And Evidence:**

I think most of the claims in this work are modestly supported:
* The best-discovered programs demonstrate a strong fit to the data, effectively capturing underlying behavioral patterns.
* The discovered programs reveal novel strategies, showcasing unique approaches to discover models of behaviors.
* The discovered programs can readily be interpreted as hypotheses about human and animal cognition, instantiating interpretable symbolic learning and decision-making algorithms.

**Essential References Not Discussed:**

N/A

**Experimental Designs Or Analyses:**

I found the baseline design in this work problematic. The authors use an RNN trained on data from all subjects collectively, rather than training separate RNNs for individual subjects, noting that the collective model generalizes poorly to held-out data. A more informative comparison might involve evaluating the best LLM-searched program against RNNs overfitted to individual subjects, as this could better highlight the strengths and limitations of each approach in capturing personalized behavioral patterns. They should also consider other variants of baselines, e.g., fine-tuning a small LM agent.

**Methods And Evaluation Criteria:**

The bandit datasets are taken from well-established work, which makes sense.

**Other Comments Or Suggestions:**

N/A

**Other Strengths And Weaknesses:**

N/A

**Questions For Authors:**

1. Why not compare the best LLM-searched program against RNNs overfitted to individual subjects?

2. How can the authors ensure that the models are not merely exploiting shortcuts in the dataset? Or are symbolic cognitive models inherently just sophisticated shortcuts anyways?

**Relation To Broader Scientific Literature:**

Computational process models have been widely utilized in computational cognitive neuroscience to study both human and animal behavior. PhD trainees in this field typically spend four to five years developing the expertise needed to construct effective models that accurately capture behavioral patterns.

The CogFunSearch workflow presented in this paper is particularly impressive, as it generates computational process model hypotheses that outperform those designed by human researchers. However, its success raises fundamental concerns about the approach itself: (1) To what extent does building computational process models contribute to our scientific understanding of cognitive behavior? (2) How valuable is it, from a scientific training perspective, for students to dedicate years to mastering a skill that large language models can now acquire with ease?

While CogFunSearch serves as a highly specialized science tool, it remains relatively narrow in scope compared to more general-purpose AI agents. Its strong performance suggests that computational process modeling may be a fundamentally "low-dimensional" task. This prompts a critical question: Can a two-step task truly capture the richness of cognitive behaviors in human and animal intelligence, especially if an LLM can model such data effortlessly?

The success of CogFunSearch implies two possible interpretations: (1) LLMs have already reached a level of intelligence sufficient to model the intelligent behaviors of other agents, or (2) the behaviors studied—such as those in the two-step task—are too low-dimensional, raising concerns about their suitability as paradigms for investigating intelligence.

I knew this paper follows an established computational neuroscience approach, and given its AI focus, I lean slightly toward acceptance. **However, the findings strongly suggest that mainstream symbolic computational neuroscience models may be oversimplifying human and animal intelligence in problematic ways.** The fact that CogFunSearch can generate superior models so easily raises fundamental concerns about whether these models genuinely capture the complexity of cognitive behavior or merely reflect an artificial, low-dimensional abstraction. If AI can so readily outperform human-designed models, it calls into question whether these models have been meaningfully advancing our understanding of intelligence—or if they have simply been reinforcing oversimplified frameworks.

**Theoretical Claims:**

There is no theoretic claim in this work.

---

> ### Author Rebuttal · Authors · 2025-03-31
>
> We thank the reviewer for their careful review of our submission, their useful comments, & are glad they found our proposed methodology “particularly impressive”.
>
> ## RNN trained on data from all subjects…rather than…separate RNNs…
>
> To clarify: This concern applies only to the human bandit dataset. We find that per-subject RNNs highly overfit to the training data & evaluated poorly on validation data, likely due to the small number of sessions & trials per subject. **We have revised the manuscript to include fits of RNNs to individual subjects’ data**, with a cross-validation setup comparable to CogFunSearch. We sweep network hidden sizes, used early stopping to mitigate overfitting, & find that the avg normalized score is 40.27$\pm$0.50, which is considerably worse than the cognitive model baseline (55.55$\pm$0.64), the best discovered program (60.93 $\pm$0.63), and the multisubject RNN (61.83$\pm$0.81). For the rat dataset, RNNs were trained separately for each subject. For the fly dataset, both RNNs and CogFunSearch programs were fit to the entire dataset, since each fly performed only one session, so it was not possible to compute held-out per-animal scores.
>
> ## does building computational process models contribute to…scientific understanding…?
>
> We thank the reviewer for raising this important philosophical question, which we agree is highlighted by our work. Computational models developed by humans reflect human-developed theories about the mechanisms at play, and are traditionally seen as tools for exploring the implications of these theories. These models can be used to make quantitative predictions about behavior in different  situations as well as different data modalities like neural recordings. They are also commonly used to understand differences in behavior, both experimentally-induced or naturally-occuring.
>
> We believe that automatically-discovered models can play a similar set of roles, so long as the theories that they express are interpretable to scientists. They play an additional role in surfacing new ideas that might be different from those that would occur to researchers, as we discuss in section 4, as well as in accelerating their discovery.
>
> ## How valuable is it…to dedicate years to mastering a skill that LLMs can now acquire with ease?
>
> We agree that as technology evolves, the set of skills that will be valuable to learn will also evolve. We see methods like ours as adding to the toolkit available, but not necessarily as reducing the skillset required of our students. Many of the skills now required in order to build and use computational models are still required to interpret & use automatically-discovered models. We agree with reviewer fWgP that CogFunSearch very much does not mean “LLMs can substitute cognitive scientists”. Instead, we hope that by facilitating discovery of interpretable quantitative models we can accelerate scientific research & enable scientists to tackle increasingly ambitious challenges.
>
> ## "Can a two-step task truly capture the richness of cognitive behaviors…?” and “the behaviors studied…are too low-dimensional…”
>
> The bandit tasks we consider strike a nice balance of not being too simple (so there’s something to learn from them) & not being too complex (so we stand a chance of learning it). Indeed, cognitive scientists have struggled with these bandit tasks for decades, despite their apparent simplicity.
>
> ## LLMs have already reached a level of intelligence sufficient to model the intelligent behaviors of other agents
>
> It is worth clarifying that currently LLMs can’t directly model this data: we are using LLMs to create variants of Python programs which serve as the cognitive models within the framework of FunSearch’s evolutionary algorithm. It is also worth noting that these LLMs are leveraging the knowledge we have built over decades of study via the seed programs provided, & exemplified in the semantically-meaningful variable names used. Reviewer fWgp phrased our goal well: “the authors did not claim "LLMs can substitute cognitive scientists" but rather focused on the specific problem of symbolic cognitive modeling, & the experiments have shown convincing results that the discovered models are of high quality.”
>
> ## “the findings suggest mainstream symbolic…models may be oversimplifying…in problematic ways.” … “If AI can so readily outperform…do these models…meaningfully advance our understanding…?”
>
> We partly agree, since RNNs (believed to achieve ceiling predictability) easily outperform previous cognitive models. Our approach largely closes this gap and arguably addresses this shortcoming. We feel that the issue is not with constructing cognitive models themselves, but rather with how we were finding them. Our results demonstrate both that better models are possible, & a novel mechanism for searching for them.
>
> ## “are models…exploiting shortcuts in datasets?”
>
> All our models were validated on held-out subjects that were never seen during training.

---

> > ### Comment · Reviewer_nJ74 · 2025-04-02
> >
> > Thanks for getting back to me. Most of my concerns are addressed. For the last part:
> >
> > > “are models…exploiting shortcuts in datasets?” All our models were validated on held-out subjects that were never seen during training.
> >
> > While validating models on held-out subjects helps mitigate certain biases, it's still possible for the entire dataset to exhibit systematic shortcuts. These shortcuts can arise from biases embedded in the data collection process, annotation conventions, or common spurious correlations that models can exploit.
> >
> > I will keep my current ratings.

---

> > > ### Author Response · Authors · 2025-04-03
> > >
> > > We agree that there may be biases embedded in the full datasets for the reasons you mention. However, this is a more general concern that is outside the scope of our work, which builds on established datasets in the field. Indeed, it is worth mentioning that these behavioral datasets have challenged neuroscience and psychology modeling efforts for decades, and while they certainly do not capture learning in every scenario, modeling behavior in this setting is a big step.
> > >
> > > Finally, we chose to run our evaluations on three datasets of different animals performing related, but different, tasks, to strengthen the claims and generality of our method.
> > >
> > > We hope this clarifies the remaining concern, and are pleased the reviewer remains supportive of our work. We encourage the reviewer to let us know if they remain concerned about this or about anything else, and we will be happy to discuss.

---

### Official Review · Reviewer_4sbb · 2025-03-13

**Overall Recommendation:** 5

**Summary:**

This paper introduces CogFunSearch, an automated approach to discovering symbolic cognitive models that accurately describe human and animal behavior. The method builds on FunSearch, a program synthesis tool powered by Large Language Models (LLMs) and an evolutionary algorithm, to systematically explore and optimize symbolic cognitive models.

**Claims And Evidence:**

The article makes three key claims:

1. The models discovered by CogFunSearch outperform state-of-the-art human-designed cognitive models in behavioral prediction across humans, rats, and fruit flies.
2. CogFunSearch can explore vast model spaces, identify solutions superior to human-designed models, and provide novel insights into cognitive mechanisms, with strong supporting evidence.
3. The discovered models remain largely interpretable.

In the conclusion, the authors state: “We find that CogFunSearch can discover programs that outperform state-of-the-art baselines for predicting animal behavior, while remaining largely interpretable.”

However, the claim regarding interpretability requires further scrutiny. The paper primarily justifies interpretability based on code readability and the intuitiveness of variable naming but does not address several critical questions:

Can researchers leverage these models to develop new methodologies?
Do these models generalize to novel datasets, or do they merely excel in fitting existing data?
Do the variable names in the high-performing models genuinely reflect their functional roles as implied by their nomenclature?

**Essential References Not Discussed:**

I did not find missing references; the citations in this paper appear to be appropriate and comprehensive.

**Experimental Designs Or Analyses:**

The experimental design of the paper is well-structured and suitable for evaluating the discovery of symbolic cognitive models. The datasets, evaluation metrics, and statistical analysis methods provide strong empirical support for the paper's key conclusions.

**Methods And Evaluation Criteria:**

In summary, the proposed methodology is well-justified, and the evaluation criteria are scientifically sound.

The ​CogFunSearch framework demonstrates ​innovative potential by enabling ​automated hypothesis discovery, thereby ​reducing reliance on manually designed cognitive models. It has achieved robust performance across three distinct behavioral datasets  while adhering to rigorous evaluation standards.

**Other Comments Or Suggestions:**

See above.

**Other Strengths And Weaknesses:**

This paper is well-structured, comprehensive, and methodologically rigorous. However, there are a few areas that could be improved to enhance its practicality and impact:

1. **Computational Efficiency Issues**: CogFunSearch requires **hundreds of thousands to millions of LLM queries** to find the optimal programs, resulting in extremely high computational costs. Although the paper mentions **rejection sampling** as an optimization technique, it lacks a detailed analysis of **computational cost vs. predictive performance gains**.

2. **Lack of Comparison with Other AI Discovery Methods**: The paper does not compare its approach with **more computationally efficient AI discovery frameworks**, such as **differentiable architecture search, Bayesian symbolic regression, or reinforcement learning-driven symbolic search**. Including such comparisons would help clarify the trade-offs between computational resource consumption and model quality.

**Questions For Authors:**

**Q1**: See Claims And Evidence

**Q2**:
Your work demonstrates the effectiveness of LLM-guided evolutionary search in discovering symbolic cognitive models. Do you believe that similar methods could be successfully applied to other domains, such as physics, chemistry, or economic modeling? If so, what characteristics of Symbolic Cognitive Models made this field particularly suitable for your approach？

**Relation To Broader Scientific Literature:**

The key contributions of this paper lie at the intersection of cognitive modeling, AI-driven scientific discovery, and program synthesis. It extends the FunSearch framework, applying LLM-guided evolutionary search to cognitive science, enabling automated symbolic model discovery. Compared to traditional methods, this approach reduces reliance on human intuition and automatically discovers models that outperform human-designed counterparts in predictive performance.

**Theoretical Claims:**

This paper does not primarily focus on formal theoretical proofs, as its main contribution lies in data-driven symbolic cognitive model discovery using LLM-powered evolutionary search. I also did not find any obviously problematic theoretical statements.

---

> ### Author Rebuttal · Authors · 2025-03-31
>
> We thank the reviewer for their careful review of our submission and their useful feedback. We are glad the reviewer found the submission “well-structured, comprehensive, and methodologically rigorous”. Below we provide responses to some of the main concerns raised.
>
> ## “Can researchers leverage these models to develop new methodologies?”
>
> Computational cognitive models for tasks of this kind are widely used for interpreting neuroscience data (do the internal variables of the model have correlates in the brain?) as well as understanding the effects of causal experiments (does damage to particular brain regions affect behavior in ways that are similar to ablation of the models?) and naturally-occurring variability (do patients with particular psychiatric diagnoses differ systematically in their model parameters?)
>
> A widely-held belief is that models which better capture behavior are more likely to be useful for these kinds of neuroscience applications. Exploring whether this is true for the models we have discovered here remains a direction for future work, but it is one we are optimistic about. This is especially true for the models discovered for the human bandit dataset, which fit data dramatically better than existing models and which contain structure qualitatively unlike the ones in those models.
>
> ## “Do these models generalize to novel datasets, or do they merely excel in fitting existing data?”
>
> In our evaluation we do cross-validation over held-out subjects. This can be considered as a form of generalization to new datasets, where the datasets come from new (held-out) animals performing the same task. Whether these models can generalize to entirely new tasks or different organisms is an interesting question for future work, but was not within the scope of our work here.
>
> ## “Do the variable names in the high-performing models genuinely reflect their functional roles as implied by their nomenclature?”
>
> Thank you for raising this question. We have manually inspected the top performing programs and found that, for the majority of cases, variable names are semantically meaningful with respect to their functional roles. We find that the meaningfulness of the variable names tends to correlate with the strength of the seed programs (i.e.starting from FullModel tends to result with more meaningful variable names versus starting from LowInfo).
>
> ## “Lack of Comparison with Other AI Discovery Methods”
>
> Our focus in this work was on whether LLM-based discovery methods could produce accurate, yet interpretable, models. We were particularly compelled to explore this due to the fact that LLMs are able to utilize the semantics of its inputs (such as variable names) to guide its generation. Some of the alternative approaches suggested would lack this semantic “understanding”.
> Nonetheless, we do agree it’s an interesting alternative to explore and will add a discussion of these ideas to our conclusions.
>
> ## “Do you believe that similar methods could be successfully applied to other domains, such as physics, chemistry, or economic modeling?”
>
> Yes! At the core of our approach is using funsearch to discover parameterized programs that can be fit to a dataset, and this can be applied to other areas of data-driven scientific discovery. One recent work in this vein is [1]. We will add a brief discussion of this extension in the final version of our manuscript.
>
> ## “what characteristics of Symbolic Cognitive Models made this field particularly suitable for your approach”
>
> It can be argued that this  field has relied heavily on models fit to data of one type (behavior) to understand data of very different types (neural activity, psychiatry, etc). This field is in crisis: deep learning has made it possible to benchmark longstanding models and find out that they just do not fit very well. It's therefore both theory-poor but also well-equipped to make use of new theories should they appear. Increasingly, it is also data-rich. These considerations make it a good place for data-driven theory discovery to have outsized impact. Further, the semantic-grounding of LLMs (discussed above) and the human-interpretable nature of Python programs allow us to leverage, and go beyond, prior cognitive models.
>
>
> [1] [Grayeli et al., 2024, “Symbolic Regression with a Learned Concept Library”](https://arxiv.org/abs/2409.09359)

---

> > ### Comment · Reviewer_4sbb · 2025-04-07
> >
> > The authors have addressed all my concerns. That supports my score. A well-done work!

---

### Decision · Program_Chairs · 2025-05-01

**Decision:**

Accept (spotlight poster)

**Comment:**

The paper introduces CogFunSearch, a novel extension of the FunSearch framework that automates the discovery of symbolic cognitive models using large language models (LLMs) combined with evolutionary algorithms and bilevel optimization. The approach is tested on behavioral datasets from humans, rats, and fruit flies, and demonstrates superior predictive performance over hand-crafted models, while preserving a degree of interpretability.

Reviewers consistently praise the clarity of the writing, the thoroughness of the evaluation, and the novelty of applying LLM-based program synthesis to cognitive science. The inclusion of detailed supplementary material and a rigorous experimental protocol further strengthens the work. However, several concerns were raised, including the high computational cost of the method, limited diversity of baseline comparisons (e.g., against better-tuned RNNs or alternative model discovery methods), and the need for a deeper discussion of the interpretability and scientific utility of the discovered models. Some reviewers also noted that the methodological novelty lies more in its application than in algorithmic innovation, and there are open questions about the broader scientific implications of automating cognitive model discovery with LLMs. Nevertheless, the author's rebuttal addresses most of the concerns. Hence, the consensus among reviewers leans clearly toward acceptance and I agree with them.